# Impacts of the 2014-2015 Holuhraun eruption on the UK atmosphere

Marsailidh M. Twigg[1], Evgenia Ilyinskaya[2*], Sonya Beccaceci[3], David C. Green[4], Matthew R. Jones[1], Ben Langford[1], Sarah R. Leeson[1], Justin J.N. Lingard[5], Gloria M. Pereira[6], Heather Carter[6]. Jan Poskitt[6], Andreas Richter[7], Stuart Ritchie[3], Ivan Simmons[1], Ron I. Smith[1], Y. Sim Tang[1], Netty Van Dijk[1], Keith Vincent[5], Eiko Nemitz[1], Massimo Vieno[1] and Christine F. Braban[1]

[1]NERC Centre for Ecology and Hydrology, Bush Estate, Penicuik, UK, EH26 0QB
[2]NERC British Geological Survey, Murchison House, W Mains Rd, Edinburgh, UK EH9 3LA
[3]Environment Division, National Physical Laboratory, Teddington, London, UK.
[4]Environmental Research Group, King's College London, Franklin-Wilkins Building, 150 Stamford Street, London, SE1 9NH
[5]Ricardo Energy & Environment, The Gemini Building, Fermi Avenue, Harwell, Didcot, OX11 0QR
[6]NERC Centre for Ecology and Hydrology, Lancaster Environment Centre, Library Avenue, Bailrigg, Lancaster, UK, LA1 4AP
[7]Institute of Environmental Physics, University of Bremen, Otto-Hahn-Allee, 1D-28359, Bremen, Germany

*now at: School of Earth and Environment, University of Leeds, Leeds, LS2 9JT, UK

*Correspondence to:* Marsailidh M. Twigg (sail@ceh.ac.uk)

## Abstract

Volcanic emissions, specifically from Iceland, pose a pan-European risk and are on the UK National Risk Register due to potential impacts on aviation, public health, agriculture, the environment and the economy, both from effusive and explosive activity. During the 2014-2015 fissure eruption at Holuhraun in Iceland, the UK atmosphere was significantly perturbed. This study focuses one major incursion in September 2014, affecting the surface concentrations of both aerosols and gases across the UK, with sites in Scotland experiencing the highest sulfur dioxide ($SO_2$) concentrations. The perturbation event observed was confirmed to originate from the fissure eruption using satellite data from GOME2B and the chemical transport model, EMEP4UK, which was used to establish the spatial distribution of the plume over the UK during the event of interest. At the two UK European Monitoring and Evaluation Program (EMEP) supersite observatories (Auchencorth Moss, SE Scotland and Harwell, SE England) significant alterations in sulfate ($SO_4^{2-}$) content of $PM_{10}$ and $PM_{2.5}$ during this event, concurrently with evidence of an increase in ultrafine aerosol, most likely due to nucleation and growth of aerosol within the plume, were observed. At Auchencorth Moss, higher hydrochloric acid (HCl) concentrations during the September event (max = 1.21 µg m$^{-3}$, c.f annual average 0.12 µg m$^{-3}$ in 2013), were assessed to be due to acid displacement of chloride (Cl$^-$) from sea salt (NaCl) to form HCl gas rather than due to primary emissions of HCl from Holuhraun. The gas and aerosol partioning at Auchencorth moss of inorganic species by thermodynamic modelling, confirmed the observed partioning of HCl. Using the data from the chemical

thermodynamic model, ISORROPIA-II, there is evidence that the background aerosol, which is typically basic at this site, became acidic with an estimated pH of 3.8 during the peak of the event.

Volcano plume episodes were periodically observed by the majority of the UK air quality monitoring networks during the first 4 months of the eruption (August – December 2014), at both hourly and monthly resolution. In the low resolution networks, which provide monthly $SO_2$ averages, concentrations were found to be significantly elevated at remote "clean" sites in NE Scotland and SW England, with record high $SO_2$ concentrations for some sites in September 2014. For sites which are regularly influenced by anthropogenic emissions, taking into account the underlying trends, the eruption led to statistically unremarkable $SO_2$ concentrations (return probabilities >0.1, ~10 months). However for a few sites, $SO_2$ concentrations were clearly much higher than has been previously observed (return probability <0.005, >3000 months). The Holuhraun Icelandic eruption has resulted in a unique study providing direct evidence of atmospheric chemistry perturbation of both gases and aerosols in the UK background atmosphere. The measurements can be used to both challenge and verify existing atmospheric chemistry of volcano plumes, especially those originating from effusive eruptions, which have been under-explored, due to limited observations available in the literature. If all European data sets were collated this would allow improved model verification and risk assessments for future volcanic eruptions of this type.

## 1    Introduction

Volcanic emissions perturb atmospheric composition in the troposphere (Bobrowski et al., 2007;Horrocks et al., 2003;Martin et al., 2008;Oppenheimer et al., 2010;Oppenheimer et al., 2006;von Glasow, 2010) via emissions of ash and/or gases and aerosols to the atmosphere, particularly during active eruptions. These emissions can directly impact humans and ecosystems (Thordarson and Self, 2003) as well as have indirect effects on climate (Gettelman et al., 2015;Schmidt et al., 2012;Schmidt et al., 2014). The injection of sulfur dioxide ($SO_2$) and sulfate ($SO_4^{2-}$), as well as aerosol in the form of ash, into the stratosphere are large events, which can have global effects, as shown by the cases of Mount Pinatubo and El Chichon where atmospheric perturbation and climate forcing were observed (Grainger and Highwood, 2003). Stratospheric perturbation generally only occurs as a result of explosive eruptions, as they have enough force to break through the tropopause to the stratosphere. In these types of eruptions, deposition of atmospheric components to the surface is diffuse and long-term.

Atmospheric chemistry in the lower troposphere and in particular the boundary layer with resultant earth surface effects are frequently associated with effusive eruptions. The 1783-84 fissure eruption of Laki is considered the "type" eruption for a long-lasting effusive eruption on Iceland. The Laki eruption dynamics and emission masses have been characterised in several papers (e.g. Thordarson and Self (2003)). In these eruptions, there is a long term flow of lava and limited ash generation with the result of a long term input of gas-phase emissions in the lower parts of the troposphere. This is contrasted with the shorter-term punctuated emissions from an explosive eruption. It is noted that eruptions maybe a mixture of both effusive and explosive, e.g. Stromboli as described by Ripepe et al. (2007). The impact of effusive eruptions on the troposphere at both local and regional scales, are most frequently studied in responsive mode, post eruption initiation. As technology and instrumentation has developed and global air quality monitoring effort has increased too, this has resulted in some cases where background conditions and the evolution of distal volcanic plumes can be now be studied.

Volcanic plumes contain elevated quantities of reactive sulfur species, primarily in the form of $SO_2$. There is still great debate whether $SO_4^{2-}$ aerosol is directly emitted from volcanos or if it is a result of rapid formation once it enters the atmosphere, which is already detectable at the crater rim (von Glasow et al., 2009). In distal plumes, it is thought that $SO_2$ will eventually form $SO_4^{2-}$ through the reaction with the OH radical, though $SO_2$ will also be removed from the troposphere through wet and dry deposition to the surface too. Quantifying the relative emission abundance of $SO_2$ and $SO_4^{2-}$ and the oxidative aging of the plume converting $SO_2$ to $SO_4^{2-}$ has been attempted previously, for example by Satsumabayashi et al. (2004) but there is a very limited number of studies (Hunton et al., 2005;Rose et al., 2006;Mather et al., 2003;Kroll et al., 2015;Boulon et al., 2011;Satsumabayashi et al., 2004) which have quantified gas and aerosol composition beyond sulfur species and provided evidence of tropospheric chemistry of distal plumes including halogen chemistry and particle growth (Boulon et al., 2011).

Previously it was thought that the chemistry of volcanic plumes in the troposphere was dominated by the oxidation of $SO_2$, however this changed when Bobrowski et al. (2003) observed in addition a relatively large emission of the bromine oxide (BrO) radical in a plume from Soufriéré Hills (von Glasow et al., 2009). It is now known that emissions of hydrochloric acid (HCl), hydrogen fluoride (HF) and hydrogen bromide (HBr) can drive the chemistry within volcanic plumes, though the presence of halogens is determined by the chemical signature of each individual volcanic system. The rates of atmospheric processing driven by halogens in volcanic plumes are of great interest, as it is thought that reaction rates may differ significantly from that observed in the background atmosphere, which is more frequently studied. This is due to higher temperatures and a unique chemical composition volcanic plumes, which is potentially able to generate radicals. It is important to study the atmospheric processing within volcanic plumes as it determines the fate and deposition of acidic compounds contained within the plume (Aiuppa et al., 2007). Studies have shown that the ratios of $SO_2$/HCl remain constant in a distal plume but the ratio decreases in the presence of clouds, as the gases dissolve in order of solubility and therefore HCl will dissolve first in the presence of clouds, though the studies mentioned were carried out within tens of kilometres from source (von Glasow et al., 2009;Burton et al., 2001;Aiuppa et al., 2007).

One aspect which is challenging for scientists is to capture both the physical characteristics and the chemical composition of volcanic plumes after mixing with the background in the distal plume, particularly at long distances away from eruption source. In the case presented the distal plume was ~ 1000 km from its source in Iceland. The primary chemical components of volcano plumes measured are $SO_2$ and $SO_4^{2-}$, however there are many other gases emitted. Other studies have in addition measured the full chemical composition of particulate matter (PM), e.g. Mather et al. (2003);Martin et al. (2008) and more recently indirect measurements of aerosol properties from satellites has been undertaken (Ebmeier et al., 2014). Aiuppa et al. (2009) and Pyle and Mather (2009) published reviews of the literature in the area of chemical degassing with a focus on emitted halogen chemicals which can occur both in the gas and aerosol phase. In most studies the ratios of HCl, HF to $SO_2$ are reported rather than the absolute concentration. Similarly the other sulfur gases, $H_2S$ and COS, are studied but less frequently. The Witham et al. (2015) report summarised the halogen acid and $H_2S$ ratios from the literature. It was noted that predominantly measurements are made close to the emission source or at a surface position downwind from the eruption either by remote sensing or direct sampling with off line analysis methods.

Only a few modelling studies (e.g.Witham et al. (2015);Schmidt et al. (2015)) and measurement approaches (Kroll et al., 2015;Businger et al., 2015) have assessed the atmospheric chemistry with an air quality impact focus. Complex perturbation of atmospheric ozone and other oxidants have been studied both in the stratosphere and troposphere, however with only limited number of observation studies. Recent reviews have indicated there are still much to be understood (Mather, 2015;von Glasow et al., 2009) to understand the input of volcano emissions and the perturbation to the atmosphere.

To date there is only one study which has demonstrated nucleation and particle growth in a distal volcanic plume. This was measured following the explosive eruption of Eyjafjallajökull in 2010, where at the GAW site Puy de Dôme, France (1,465 m above sea level), nucleation and secondary aerosol formation events within the volcanic plume were observed (Boulon et al., 2011). The site however was in the free troposphere and occurred in a plume which was ash rich. To date there is very limited evidence of the processes of particle nucleation and secondary aerosol formation in distal plumes from effusive eruptions.

The recent Holuhraun eruption within the Bárðarbunga  volcanic system in Iceland (August 2014 - February 2015) was the largest Icelandic eruption in terms of erupted magma and gas volume since the 1783-1784 CE Laki event, producing 1.6 km$^3$ of lava and total $SO_2$ emissions of 11±5 Mt during a period of 6 months (Gíslason et al., 2015). It was almost purely effusive, hence producing negligible amounts of ash, but repeatedly causing severe air pollution events in populated areas of Iceland due to high gas and aerosol concentrations. The ground level concentration of $SO_2$ exceeded the hourly health limit (350 µg m$^{-3}$) over much of the country for periods of up to several weeks (Gíslason et al., 2015) and there were complaints as far as Scandinavia of a foul smell, which has been attributed to sulfurous compounds from the fissure eruption using satellite data (Grahn et al., 2015). Exceedances in the hourly health limits were also observed for $SO_2$ periodically in Northern Finland at surface observation sites, which were confirmed by satellite observations (Ialongo et al., 2015). In Europe, anthropogenic emissions of sulfur have been declining over the past few decades and hence lower concentrations are observed widely (Fowler et al., 2007). The 28 countries European Union member countries (EU-28) total annual emissions of sulfur oxides in 2010 and 2011 were ~4.6 Mt (http://www.eea.europa.eu/data-and-maps/daviz/emission-trends-of-sulphur-oxides#tab-chart_1) and therefore the Holuhraun volcanic eruption added more than twice the EU-28 annual sulfur emissions to the atmosphere in just six months (Schmidt et al., 2015). This eruption provided the unique opportunity in Europe to study the impact of a large point source $SO_2$ emission.

This paper studies the volcanic impact on the UK atmosphere and focuses on one major incursion in September 2014 during the Holuhraun eruption and provides the first evidence of wide scale effects, based on the measurements from the UK air quality monitoring networks which deliver data at both high (hourly) and low (monthly) temporal resolution. These observations provide information on the chemical composition of the distal plume, ~ 1000 km downwind of Iceland. Because Icelandic air arrives at the UK on northerly trajectories, the background air is clean and there is little interference from anthropogenic emissions when the air arrives at the northern UK.

In 2014, hourly resolution measurements of $SO_2$ were made by the UK Automatic and Rural Monitoring Network (AURN, http://uk-air.defra.gov.uk/networks/network-info?view=aurn) and by the two UK European

Monitoring and Evaluation Program (EMEP) (Torseth et al., 2012) atmospheric observatories (Harwell, SE England, UK and Auchencorth Moss, SE Scotland, UK), which also form part of the ACTRIS Infrastructure Network (http://www.actris.eu/). Additional high resolution physical particulate matter (PM) mass and size distribution (refer to section 2.2), as well as chemical composition measurements (refer to section 2.1) at the two EMEP observatories are presented. In addition supplementary evidence of long term perturbations in the UK background at a lower resolution during the volcanic event from the UK Acid Gas and Aerosol NETwork (AGANET) and Precipitation network (Precip-Net) are highlighted (refer to section 2.3).

## 2        Methods

### 2.1        Basics of MARGA operation

The Measurement of Aerosols and Reactive Gases Analyser (MARGA, Metrohm Applikon B.V, NL) provides hourly resolution measurements of water soluble inogranic aerosol speciation ( $SO_4^{2-}$, $Cl^-$, $NO_3^-$, $NH_4^+$, $Na^+$, $K^+$, $Ca^{2+}$ and $Mg^{2+}$) and gases ($SO_2$, $HCl$, $HNO_3$, $HONO$ and $NH_3$). At the two field sites Harwell and Auchencorth Moss (Figure 1), the instruments are configured to have two sample boxes, one for $PM_{10}$ and one for $PM_{2.5}$. The instruments use wet rotating denuders (WRD) (Wyers et al., 1993) and steam jet aerosol collectors (SJAC) (Khlystov et al., 1995) for sampling of gases and aerosols respectively. Analysis is carried out online by ion chromatography (both anion and cation) at an hourly resolution. A detailed description of the instrument and quality assurance/ quality control (QA/QC) procedures used by both instruments is given in Twigg et al. (2015). There is one operational difference between the Auchencorth and Harwell instruments, where Auchencorth Moss uses preconcentration columns (Metrosep A PCC 1 HC ion chromatography (IC) preconcentration column (2.29 mL) for anions and a Metrosep C PCC1 HC IC pre-concentration column (3.21 mL) for cations) on the IC to achieve a lower detection limit (DL) compared to the Harwell instrument which uses fixed loops (250μL for anions and 480μL for cations) and therefore has an order of magnitude higher DL as described by Makkonen et al. (2012). Data from both MARGA instruments are available in the UK-Air (http://uk-air.defra.gov.uk/data/) and EBAS (http://ebas.nilu.no/default.aspx) databases.

### 2.2        SMPS

At both EMEP supersites scanning mobility particle sizers (SMPS) are installed which count the individual aerosol numbers within predefined size bins of aerosols. At Harwell, aerosol number size distributions measured in the range of 16.55 to 604.3 nm by the SMPS (Electrostatic classifier 3080, differential mobility analyser 3081, and condensation particle counter 3775, all TSI Inc.). The system sampled air at 4 m above ground level, through a $PM_1$ cyclone before entering the analyser via a drier which ensured the relative humidity of the sample air was kept below 45%. The aerosol sample flow rate was set to 0.3 L min$^{-1}$ and the Classifier sheath flow was maintained at 3 L min$^{-1}$; a detailed description of the method and set-up employed at Harwell can be found in Beccaceci et al. (2013) and data is freely available through the UK-Air website.

At Auchencorth Moss aerosol size distributions in the range of 14-673 nm were set to be measured using the SMPS (Electrostatic classifier 3081, differential mobility analyser 3080 and condensation particle counter 3775, all TSI, Inc.). Air was sampled at 2 m above ground level through a $PM_{10}$ head and $PM_{2.5}$ cyclone before

entering the analyser via a drier which ensured the relative humidity of the sample air was kept below 45%. The aerosol sample flow rate was set to 0.3 L min$^{-1}$ and the classifier sheath flow was maintained at 3 L min$^{-1}$ as set out inWiedensohler et al. (2012). In October 2015, the Auchencorth Moss SMPS took part in an intercomparison organised by the EU Horizon 2020 ACTRIS 2 (aerosol, clouds and trace gases research infrastructure), held at the world aerosol calibration centre (TROPOS, Leipzig, Germany). During this exercise the classifier used at Auchencorth was found to have an offset and was starting a scan at 35 nm instead of 14 nm, though it is unclear if this may have slowly drifted over the 18 months since installation at the site. Therefore data presented from Auchencorth Moss is a qualitative indicator of an increase in ultrafine particles as the size distribution could not be verified.

## 2.3 AGANet DELTA and Precip-Net

The DEnuder for Long-Term Atmospheric sampling (DELTA), used in AGANet across the UK, is described by Sutton et al. (2001), is used to measure the spatial concentrations of both inorganic trace gases (NH$_3$, HNO$_3$, SO$_2$ and HCl) and counterpart aerosols at a monthly resolution across the UK. The sampling system consists of a series of coated denuders (to capture gases) and filters (to capture the aerosol). Air is sampled at a flowrate of 0.2 -0.4 L min$^{-1}$, with the sampling inlet at a height of 1.5 m. The first pair of denuders (15 cm) after the inlet are coated with K$_2$CO$_3$/glycerol to capture acidic gases. The next pair of denuders are coated with citric acid to capture gaseous NH$_3$. A filter pack is situated at the end of the sampling train, containing two cellulose coated filters: the first is impregnated with K$_2$CO$_3$ to capture and retain NO$_3^-$, SO$_4^{2-}$, Cl$^-$ and Na$^+$, Ca$^{2+}$ and Mg$^{2+}$ aerosol. The second filter is impregnated with citric acid to capture NH$_4^+$. Downstream of the sampling train is a gas meter, to record the volume of air sampled and an air pump. A DELTA sampling train is exposed for 1 month and samplers are extracted with deionised water. Chemical analysis is performed by ion chromatography and flow injection analysis, further details of both the sampling method and analytical analysis are contained in Tang et al. (2009). The monitoring sites in AGANet are highlighted in Figure 1. The wet deposition of pollutants in the UK is monitored within Precip-Net. Precip-Net uses bulk precipitation samplers at 39 non-urban sites with fortnightly sample collection. Samples are analysed for cations (Na$^+$, Ca$^{2+}$, Mg$^{2+}$, K$^+$, NH$_4^+$) and anions (PO$_4^{3-}$, NO$_3^-$, SO$_4^{2-}$, Cl$^-$) using ion chromatography (further details of both the sample method and analysis can be found in Irwin et al. (2002)). Data from both AGANet and Percip-Net are freely available from UK-Air.

## 2.4 GOME2 Volcanic SO$_2$ detection.

The GOME2 instrument on MetOp-B is a nadir viewing UV/visible spectrometer with a spatial resolution of 40 x 80 km$^2$. SO$_2$ column densities are retrieved using a Differential Optical Absorption Spectroscopy approach including a non-linearity correction for SO$_2$ saturation effects (Richter, 2009). Satellite UV/vis retrievals yield integrated vertical columns of absorbing species and usually do not provide information about the vertical distribution of a trace gas. As the sensitivity of the observations decreases towards the surface, an assumption has to be made in the retrieval on the vertical profile of the target species in order to apply appropriate weights called air mass factors. Here, the standard volcanic product from the University of Bremen is used (http://www.iup.uni-bremen.de/doas/gome2_so2_alert.htm) which assumes a volcanic eruption profile with an

$SO_2$ peak at 10 km height. As no corrections are made for the effects of deviations from the assumed plume height of 10 km, the data shown should be used as qualitative indicator only.

## 2.5    EMEP4UK chemical transport model

The EMEP4UK model rv4.3 (Vieno et al., 2016), is a chemical transport model which is the regional application of the EMEP MSC-W model (Simpson et al., 2012), which is used in this study to identify and investigate the spatial distribution of the volcano plume. The meteorological driver used in the EMEP4UK model is the weather and research and forecast model (WRF) version 3.6. More details of the model description and setup of the model can be found in Vieno et al. (2014) and Vieno et al. (2010). The model domain include all Europe and part of Russia with a horizontal resolution of 50 km x 50 km, with anthropogenic and biogenic emissions included based on the emissions the year 2012.  The specific Icelandic volcano emissions in the run were set to 680 kg/s (Gíslason et al., 2015) from August 31st 2014 to the end of 2014, with the period of the 19 September 2014 to 24 September 2014 presented in this study. The emissions are injected into the model vertical column equally from the 1km to 3 km.

## 2.6    ISORROPIA thermodynamic model

The chemical thermodynamics model, ISORROPIA II (Fountoukis and Nenes, 2007), is used to determine the theoretical chemical composition based on the gas/aerosol equilibrium partitioning of the inorganic species measured by the MARGA instrument at Auchencorth Moss. The model was run using as an input the bulk (i.e. gas + aerosol) concentration of all compounds (ammonium, nitrate, sulfate and chloride) measured by the MARGA (input in $\mu$mol m$^{-3}$) with measured temperature and relative humidity and operated in the metastable, forward reaction. The model was used to establish if the observed gaseous concentrations could be explained solely by the thermodynamic equilibrium of the observed species, as there is very little evidence in the literature of direct acid displacement.

## 2.7    Statistical analysis of AGA-Net data

As well as high temporal resolution analysis (hourly measurements) of the volcanic plume, the monthly average $SO_2$ from AGANet measurements (Section 2.3) were analysed to assess the impact of the fissure eruption on the background atmosphere across the UK. The plume was clearly observable in this dataset. In order to understand the relative magnitude of the perturbation on a national scale for both air quality and sulfur deposition, against a background of decreasing anthropogenic emissions a statistical treatment was thought to be appropriate. Specifically the likelihood of a reoccurrence of the observed concentrations in the UK background was calculated as many AGANet sites show decreasing trends over time for $SO_2$ and $SO_4^{2-}$, reflecting decreases in emissions. This is observed for both the annual mean concentration and the annual maximum concentration. A high concentration superimposed on a downward concentration trend in a measurement series over a period appears to be a less unlikely observation at the end of the time period than at the beginning. Therefore the AGANet measurement data were adjusted to remove any underlying trends before further analysis to assess the unusualness of the September 2014 elevated $SO_2$ concentrations. Theoretically exceedances over a threshold follow a Pareto distribution and the threshold was chosen by fitting an 85% quartile regression to this dataset

using a smoothing spline for each site individually (i.e. there was no assumption of a general trend). This methodology has previously been developed much further for application to case studies with substantially more data (Chavez-Demoulin and Davison, 2005;Northrop and Jonathan, 2011;Reich et al., 2011). The fitted Pareto distribution generates the probabilities of occurrence of the elevated concentrations − which in this case were associated with the volcanic plume. This is expressed in Table 1 as a return probability and return time, which is the statistical likelihood of a similar concentration to be observed again based on the long term trend of $SO_2$ over the 1999-2014 period at each site expressed in the resolution of the measurements.

## 3    Results and Discussion

### 3.1    Identification of a volcanic plume in the UK atmosphere

During the Holuhraun eruption, the volcanic plume passed periodically over the UK, with a major event occurring between September 21st and 23rd September 2014. This plume was first detected at the UK supersite in Scotland at Auchencorth Moss at 12:00 (GMT), followed by Harwell in England at 15:00 (GMT) (Figure 2). The plume moved across the UK (with the exception of parts of Northern Ireland) and Automatic Urban and Rural Network (AURN) $SO_2$ observations at selected sites are summarised in Figure 2. Scotland (Dundee, Croy and Auchencorth) observed higher concentrations of $SO_2$ compared to the rest of UK. The sites in Southern Scotland (Auchencorth Moss and Croy), however, were only exposed to the main plume on the 21st September whereas the event affected the rest of the UK intermittently for the next 72 hours (Figure 2). The peak $SO_2$ concentration measured by the MARGA at Auchencorth was 66.8 µg m$^{-3}$ (Figure 2) compared with the annual average of $SO_2$ of 0.14 µg m$^{-3}$ in 2013 at the site. It has to be noted that the $SO_2$ concentration at Auchencorth Moss was underestimated between 11:00 and 22:00 on the 21 September 2014, because the standard instrument configuration was optimised for < 1 µg m$^{-3}$ detection. The maximum reported $SO_2$ concentration during the event at Harwell reported by the MARGA was lower, peaking at 45.7 µg m$^{-3}$ (annual average concentration in 2013 was 0.46 µg m$^{-3}$) occurring on the 22nd September. Although $SO_2$ concentrations were elevated in many parts of the UK, they were notably below the 24 hour-average air quality limit of 125 µg m$^{-3}$ set under the EU Air Quality Framework Directive (Directive 2008/50/EC). The $SO_2$ plume was also observed across Ireland, Netherlands, Belgium Finland and Austria (TS-2 in Supplementary Material of Gíslason et al. (2015) and Ialongo et al. (2015)) during different periods of the fissure eruption.

Supporting evidence that the ground-based measurements in September 2014 were picking up a volcanic signal is provided by the GOME2 instrument on the MetOp-B satellite, as it was able to track the $SO_2$ plume from the Holuhraun eruption site to the UK (Figure 3) on the 20th and 21st September. While satellite observations only provide tropospheric columns of $SO_2$ and not the concentrations at surface which are reported by the ground-based measurements, they do provide important information on the origin of enhanced $SO_2$ values. As the sensitivity of satellite retrievals to surface $SO_2$ is low, sulfur dioxide from surface pollution can rarely be detected over Europe. Volcanic plumes in contrast can readily be observed in the data as they usually extend to higher altitudes and contain much larger vertical columns of $SO_2$. The fact that several Dobson Units of $SO_2$ are observed in GOME2 data on September 20[th] and 21[st], 2014 over the UK, and that the $SO_2$ plume originates from Iceland strongly indicates that the observed $SO_2$ is of volcanic origin. It is therefore very probably that

simultaneous $SO_2$ enhancements measured at the surface are also linked to the volcanic emissions. Modelling of the plume by EMEP4UK further confirmed the volcanic origin and dispersion of the observed $SO_2$ plumes both at Harwell and Auchencorth Moss (Figure 4), as did back trajectories using the HYSPLIT model, which can be found in figures S1-S2 in the supplementary material. To provide evidence that the EMEP4UK model was able to replicate the spatial distribution of the plume a comparison of the time series of observations to the model at both sites is given the supplementary material (figure S3). Though in general the EMEP4UK model does get the spatial distribution of the plume correct there are number of explanations why the model does not precisely replicate the plume compared to the measurements at the surface. The reasons include there a number of input variables which would have impacted the distribution of the plume including injection height, daily emission rates at source and oxidation rate within the plume, in addition the model output is 50 Km x 50 Km resolution.

### 3.2    Chemistry within the volcanic plume

### 3.2.1    Formation of sulfate aerosols

Current understanding of volcanic emissions is that the major fraction of observed $SO_4^{2-}$ is not directly emitted from the magma but is formed as secondary aerosol through oxidation of $SO_2$ in the atmosphere (Mather et al., 2013), though there are some reports suggesting primary emissions are possible (Allen et al., 2002;Zelenski et al., 2015). As shown in Figure 2, both atmospheric observatories in the UK detected an increase in $SO_4^{2-}$ during the volcanic plume event in September 2014. $SO_2$ oxidation with the hydroxyl radical in the troposphere can be slow, taking up to two weeks under some conditions, though if $SO_2$ is taken up onto particles, oxidation rates are much faster, resulting in a lifetime of days or hours in clouds as $SO_2$ (von Glasow et al., 2009). In order to understand the oxidation of an $SO_2$ plume, Satsumabayashi et al. (2004) defined a sulfur conversion ratio ($F_s$) as $F_s = [PM_{2.5} \ SO_4^{2-}]/([SO_2]+ [PM_{2.5} \ SO_4^{2-}])$ (all concentrations in µg S m$^{-3}$), where a smaller value suggests a plume which has not undergone considerable atmospheric processing. The UK observatory datasets showed $F_s$ decreasing from ~1 (all S in the form of $SO_4^{2-}$) to $F_s$~0.2, (Figure 5) during the event implying that $SO_2$ oxidation had not had sufficient residence time (and oxidant exposure) to be complete. As the plume passed over the sites, the presence of ultrafine particles also became pronounced, compared to the background atmosphere in the previous 24 hours (Figure 5). The volcano plume event was characterised by the high particle number density at low diameters, increasing in diameter with time (initiating at ~1200 hours GMT on the 21st, Figure 5). The feature of increasing particle numbers, or "banana" shape in figure 5, starting with high particle numbers at the detection limit of the SMPS is characteristic of particle nucleation and growth; however, as it is not a Lagrangian measurement, and because the nucleation does not represent a wide-spread regional phenomenon (as it probably does, e.g., in the nucleation studies conducted in the Boreal environment; (Kulmala et al., 1998)), the evolution of the size distribution with time needs to be interpreted with caution: only if trajectories and wind speed do not change with time can the temporal change at the fixed site be translated into the temporal change within the plume. It is possible that a population of ultrafine $H_2SO_4$ particles were emitted or formed at source, however, it is highly unlikely due to the transport time, refer to the HYSPLIT back trajectories found in figure S1, that aerosol would have remained in the ultrafine fraction observed as they would have undergone further growth by coagulation and further condensation of condensable vapours. It is much more likely that, sulfuric acid was formed during transport through oxidation of the high concentrations of $SO_2$ by the OH radical the production of

which is linked to solar radiation. It is hypothesized that with increasing time after sunrise, the measurements at Auchencorth reflect particles whose nucleation was initiated further and further away from Auchencorth and had increasing time to grow during transport. The SMPS at Harwell also recorded similar events as the plume passed over. This is the first evidence of boundary layer surface-level particle growth observations in a distal volcanic plume for the UK and complements observations from the 2010, which is the only previous report of nucleation and secondary aerosol formation event reported for a distal plume during the explosive, ash-rich plume (Eyjafjallajökull in 2010) at an elevated free tropospheric atmospheric station in Europe (Puy de Dôme observatory, France) (Boulon et al., 2011). At that station, the free tropospheric conditions and size range of measurements allowed the clear interpretation of particle nucleation. Though nucleation is thought to occur in the free troposphere where there is a low particle surface area there is increasing evidence in the literature of nucleation events measured at the surface in polluted environments: Hamed et al. (2007) observed nucleation events in the Po Valley, Italy, and Kulmala et al. (2005) in New Delhi. In addition, we cannot rule out that other compounds are taking part in the nucleation process and aerosol growth observed. Furthermore as well as the particle population changes observed, the measurement indicates that there was an air quality impact from particulates during the 2014 eruption in addition to the $SO_2$ air quality impacts discussed in the recent study of Schmidt et al. (2015).

### 3.2.2    Modification of the chemical composition within a volcanic plume

The chemical composition of $PM_{2.5}$ and the gas concentrations observed during the event at Auchencorth are summarised in Figure 6. It is clear that the aerosol was dominated by $SO_4^{2-}$. Whilst the aerosol at this site is normally basic, with free ammonia ($NH_3$) available based on ion balance studies (Twigg et al., 2015), during the plume event there is evidence that the aerosol turned acidic. The aerosol pH was confirmed using the results of [$H^+$] calculated from ISORROPIA-II model to calculate pH. During the plume event it was found that the pH dropped from pH 7.97 at 09:00 GMT on the 21/09/14 to pH 3.80 at 15:00 on the 21/09/14.

During the event the measurements at the background site clearly showed that there was an increase not only in the sulfur species but also in hydrochloric acid gas (HCl) and a variety of other chemical species in both gas and aerosol phase (Figure 6). HCl peaked at 1.21 μg m$^{-3}$ during the event compared with an annual average of 0.12 μg m$^{-3}$ in 2013. As discussed in Aiuppa (2009), Pyle and Mather (2009) and summarised in Witham et al. (2015) and the literature cited therein, primary emissions of HCl from volcanoes can vary enormously depending on the magma type and the particular eruption characteristics. The near-source measurements of the gas composition from the Holuhraun eruption indicated that the gas phase in the plume was proportionally very low in halogen content, with a molar $HCl/SO_2$ ratio of <1% (Burton et al., 2015). It is unlikely that HCl would persist longer in a plume than $SO_2$ given the high solubility of HCl and comparably low reactivity of $SO_2$. However, given that the $SO_4^{2-}$ aerosol is highly acidic, the HCl would need to be scavenged onto other non-sulfate aerosol or into cloud droplets. Hence the elevated HCl observed in the plume event is either due to transport of primary HCl or displacement of HCl from background sea salt aerosol or a combination of the two. It is hypothesised that the most likely explanation for the observation of HCl coinciding with the plume is the oxidation of $SO_2$ to sulfuric acid which then displaced Cl$^-$ in pre-existing sea salt aerosol (NaCl) in the air mass. The thermodynamic model ISORROPIA-II (Fountoukis and Nenes, 2007) was used to calculate the theoretical partitioning between the gas

and aerosol phase. The model clearly reproduces the HCl peak which is attributed to the displacement of $Cl^-$ from sea salt (Figure 6). Further evidence of acid displacement was found at Auchencorth Moss when the ratio of $Na^+$ and $Cl^-$ was compared to the known ratio of sea water, where a large relative depletion of aerosol $Cl^-$ was found during elevated $SO_4^{2-}$, represented by a change in colour of the markers in Figure 7. This is not the first time the site has observed acid displacement, Twigg *et al.* (2015) observed the site to be rich in sea salt due to its proximity to the sea, with 35% of the annual average of the inorganic composition of $PM_{2.5}$ attributed to sea salt. During high nitrate ($NO_3^-$) episodes it was observed on occasions that this coincided with an apparent depletion of $Cl^-$ from sea salt, which was attributed to the displacement of $Cl^-$ by $HNO_3$. Roberts et al. (2009) has previously modelled the likely perturbations within a volcano plume upon mixing with background air, which found that both ozone and $NO_x$ were perturbed with bromine chemistry as the key driver. In order understand the net impact of this eruption on the chemical perturbation of background sea salt aerosol and the production of HCl in the gas phase, there would need to be a full chemical transport model which includes both chloride and bromine chemistry, which is beyond the scope of this paper. It is however noted, that between 09:00 (GMT) on 21/09 and 03:00 (GMT) on 22/09, the $Na^+$ was known to be underestimated, attributed to acidic composition of the aerosol resulting in a reduction in the performance of the cation column (concentration of the $Li^+$ internal standard decreased). Whilst correction based on the $Li^+$ standard is possible, this assumes that the retention was similarly depressed for all cations. The data therefore have been flagged as invalid during the QA/QC procedures of data submission to UK-Air and EMEP but have been presented here as it is thought to be useful data for research purposes. As such the depletion of $Cl^-$ is thought to be even greater than that demonstrated in Figure 7.

It is surprising that both acid displacement and nucleation events would be observed in parallel, as one process would be expected to be favoured over the other, depending on the aerosol surface area. Instead it is hypothesized that though both events are observed at the same time, it does not necessarily mean that they occur at the same time and at the same location. For the displacement reaction to occur, the sea spray aerosol needs to mix with the volcanic plume from aloft. It is feasible that this mixing generates 'pockets of air' that are dominated by the volcanically influenced air (where nucleation is favoured) and others which have efficiently mixed with the sea salt (where heterogeneous chemistry is favoured). Through continued mixing and through the long-term integration of the measurement both would be reported at the same time. Similarly, the conditions that favour one process over the other may vary along the trajectory between Iceland and the UK.

### 3.3    Long term perturbation of the UK atmosphere

The relative importance of the volcanic plume over the four months on the UK surface composition and the wider region with respect to air quality and acid deposition can only be assessed with measurements over a wider geographic region. The low-temporal resolution (monthly) measurements of gas and aerosol composition from AGANet at 30 sites (Figure 1) provided a clear signal of the impact across the UK in particular for $SO_2$ (Figure 8). The national average concentration of $SO_2$ from this network for September 2014 was about a factor of six larger than in the preceding month. Remote sites such as Strathvaich Dam in northern Scotland (Figure 8: middle panel), which typically experience very little anthropogenic air pollution, experienced the highest monthly $SO_2$ concentration on record (network operational since 1999), with September and October concentrations an order of magnitude higher than the long-term average (2 µg m$^{-3}$ c.f. 0.2 µg m$^{-3}$). Similarly Yarner Wood in the south

west of England experienced the highest concentrations on record for the site, and even taking into account the underlying decreasing trend in $SO_2$ concentrations, return probabilities were as low as $3\times10^{-4}$ (Table 1, refer to section 2.7 for statistical methods). At sites which are much more anthropogenically influenced, though the plume is clearly observable, the $SO_2$ concentration is unremarkable with return probabilities of $>1\times10^{-1}$ (e.g. Detling). A few sites on the Western side of the UK were not in the pathway of the plume therefore no elevated concentrations were observed, e.g. Rum. When assessing the wet deposition from Precip-Net, it was seen that many sites across the UK did experience elevated $SO_4^{2-}$ concentrations in rain in September and October 2014 (Figure 9 upper panel). Again, at particular sites in northern Scotland and South West England elevated concentrations were observed, whereas in Northern Ireland and parts of Wales no increase in $SO_4^{2-}$ concentrations was evident. It has to be noted, however that there was exceptionally low rainfall during September 2014 across the UK, with the month being the driest on record for the UK, based on a series from 1910, (which also equalled fifth driest in the England & Wales Precipitation series from 1766) (Parry et al., 2014). The majority of the western UK received less than 20% of the long‑term average rainfall, hence the amount of sulfur deposited by wet deposition during this period was not important to the UK annual sulfur deposition budget and hence the environmental impact through acid deposition will have been minimal (Figure 9). It therefore has to be noted that the reported high $SO_4^{2-}$ could be the result of lack of dilution due to low precipitation and cannot be directly attributed to the volcanic plume.

## 4    Conclusions

The Holuhraun eruption perturbed the UK atmospheric composition periodically during the latter part of 2014. Elevated $SO_2$ was observed by the networks at both high and low resolution. These observations complement the study by Schmidt et al. (2015) who reported similar observations for $SO_2$ across Europe for the same period. This study, however, provides further details of the chemistry within the volcanic plume which are not addressed by Schmidt et al. (2015). In this study high $SO_2$ concentrations, were demonstrated to have resulted in an increase in tropospheric HCl due to the acid displacement of $Cl^-$ from sea salt at the EMEP supersite Auchencorth Moss. Elevated particulate $SO_4^{-2}$ and particle size distributions from the two EMEP supersites provide observational evidence for new particle formation and growth was occurring as the plume passed over the UK. Future work now needs to be done investigate the direct and indirect effects of the perturbation of chemistry, specifically with regards to human health and crop yields.

The analysis also provides evidence to support the recent modelling work undertaken which concluded that volcano eruptions in Iceland will intermittently affect the UK (Witham et al., 2015) with the effects varying both spatially and temporally during an eruption, primarily driven by meteorology. There is a significant difference in effects on both human health and ecosystem effects between acidic–non-acid aerosol and this study presents the first evidence that volcanic aerosol reaching the UK can be acidic, however this will be highly dependent on the mixing of the plume with the background atmosphere. There are also further impacts which have not yet been fully assessed, for example the net effect on climate (Gauci et al., 2008;Gettelman et al., 2015) and ecosystem function.

The study has highlighted that even though anthropogenic $SO_2$ concentrations have dramatically decreased in the UK over the last 30 years, there is still a need to maintain the network of analysers as it is not just needed to confirm recovery, but also provides a useful tool to track the progression and impact of volcanic plumes and other pollution events. High resolution chemical composition of aerosol is essential for the identification of the origin of aerosol events observed concurrently with $SO_2$ plumes and to understand the atmospheric chemistry. This paper presents the first detailed observations of chemistry within a distal volcano plume at the surface in the UK. This dataset is unique and can be used by modellers to test long term impacts of volcanic eruptions and the evolution of the plume chemistry.

While the 2014-2015 eruption in Holuhraun system was the largest eruption in Europe in over 200 years, there is a potential for even larger events. For example, the 1783-84 Laki eruption was over 10 times larger in terms of erupted magma and gas volume. An event of this magnitude would cause significant and wide-spread pollution over Europe and even cause excess mortality (Schmidt et al., 2011). Though some work has been done on a limited dataset of the European air quality networks by Schmidt et al. (2015) and Gíslason et al. (2015), a further study is required of the data from across the European compliance networks, as well as the EMEP and ACTRIS networks to integrate both particle characterisation and gas chemical composition. This would allow the Holuhraun event to be fully characterised and quantified.

**Acknowledgements**

The authors would like to thank Dr Susan Loughlin at the British Geological Survey for her encouragement and useful discussion on this work. We would also like to thank the late Prof Roland Von Glasow for early discussions on the data presented here, and acknowledge his great contribution to the study of halogens in the troposphere. The authors would like to acknowledge the following for the funding of the UK measurement networks, the UK Department for Environment, Food and Rural Affairs (Defra) and the Devolved Administrations, through the Operation and Management of the EMEP Supersite, the UK Eutrophying and Acidifying Atmospheric Pollutants (UKEAP) project (AQ0647), the UK particle number and concentrations network, as well as the UK Automatic Urban and Rural Network, as well as FP7 ACTRIS and Horizon 2020 ACTRIS-2. E. Ilyinskaya work was funded by NERC Urgency Grant NE/M021130/1 and the European Community under the FP7 Grant Agreement No. 308377 (Project FUTUREVOLC).

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

**Table 1. Statistical analysis of the UK AGANET sites SO₂ (refer to Section 2.7 for the methods), where the return probability, is the statistical likelihood of a similar concentration to be observed again based on the long term trend of SO₂ at each site. (Refer to Figure 1 for site locations).**

| Site | September 2014 average SO$_2$ ($\mu g\ m^{-3}$) | Fitted 85% quartile | residual | Return probability | Return period (months) |
|---|---|---|---|---|---|
| Yarner Wood | 3.48 | 0.40 | 3.08 | $3.11 \times 10^{-4}$ | 3218.5 |
| Rothamsted | 2.98 | 0.78 | 2.20 | $9.00 \times 10^{-03}$ | 111.1 |
| London Cromwell Road | 2.37 | 0.54 | 1.83 | $6.90 \times 10^{-03}$ | 144.9 |
| Ladybower | 2.29 | 1.34 | 0.95 | $3.22 \times 10^{-02}$ | 31.0 |
| Harwell | 2.28 | 0.82 | 1.46 | $6.73 \times 10^{-03}$ | 148.7 |
| Halladale | 2.11 | 1.03 | 1.08 | $5.60 \times 10^{-03}$ | 178.6 |
| Strathvaich | 2.09 | 0.28 | 1.81 | $2.40 \times 10^{-03}$ | 417.4 |
| Shetland | 2.04 | 1.02 | 1.02 | $6.20 \times 10^{-03}$ | 161.2 |
| Auchencorth Moss | 2.00 | 0.37 | 1.63 | $8.71 \times 10^{-03}$ | 114.8 |
| Glensaugh | 1.95 | 0.57 | 1.38 | $6.17 \times 10^{-03}$ | 162.0 |
| Stoke Ferry | 1.84 | 0.68 | 1.16 | $1.70 \times 10^{-02}$ | 58.8 |
| Sutton Bonnington | 1.70 | 1.11 | 0.59 | $7.11 \times 10^{-02}$ | 14.1 |
| Barcombe Mills | 1.68 | 0.74 | 0.94 | $8.98 \times 10^{-03}$ | 111.3 |
| High Muffles | 1.67 | 1.14 | 0.53 | $7.72 \times 10^{-02}$ | 12.9 |
| Lagganlia | 1.65 | 1.14 | 0.51 | $7.28 \times 10^{-03}$ | 137.4 |
| Eskdalemuir | 1.45 | 0.42 | 1.03 | $5.65 \times 10^{-03}$ | 177.1 |
| Bush Estate | 1.38 | 0.60 | 0.78 | $4.55 \times 10^{-02}$ | 22.0 |
| Moorhouse | 1.24 | 0.47 | 0.77 | $1.10 \times 10^{-02}$ | 90.6 |
| Narberth | 1.20 | 1.08 | 0.12 | $8.69 \times 10^{-02}$ | 11.5 |
| Rosemaund | 1.15 | 0.51 | 0.64 | $1.19 \times 10^{-02}$ | 84.0 |
| Cwmystwyth | 1.10 | 0.49 | 0.61 | $2.11 \times 10^{-02}$ | 47.3 |
| Plas Y Brenin | 1.08 | 1.08 | 0.00 | $1.41 \times 10^{-01}$ | 7.1 |
| Caenby | 0.98 | 0.98 | 0.00 | $1.44 \times 10^{-01}$ | 6.9 |
| Edinburgh St Leonards | 0.61 | 1.32 | -0.71 | $4.06 \times 10^{-01}$ | 2.5 |
| Hillsborough | 0.61 | 0.61 | 0.00 | $1.37 \times 10^{-01}$ | 7.3 |
| Detling | 0.58 | 1.01 | -0.43 | $6.79 \times 10^{-01}$ | 1.5 |
| Goonhilly | 0.47 | 0.49 | -0.02 | $1.36 \times 10^{-01}$ | 7.4 |
| Lough Navar | 0.39 | 0.39 | 0.00 | $1.75 \times 10^{-01}$ | 5.7 |
| Rum | 0.07 | 0.21 | -0.14 | $9.08 \times 10^{-01}$ | 1.1 |
| Carradale | nd | | | | |

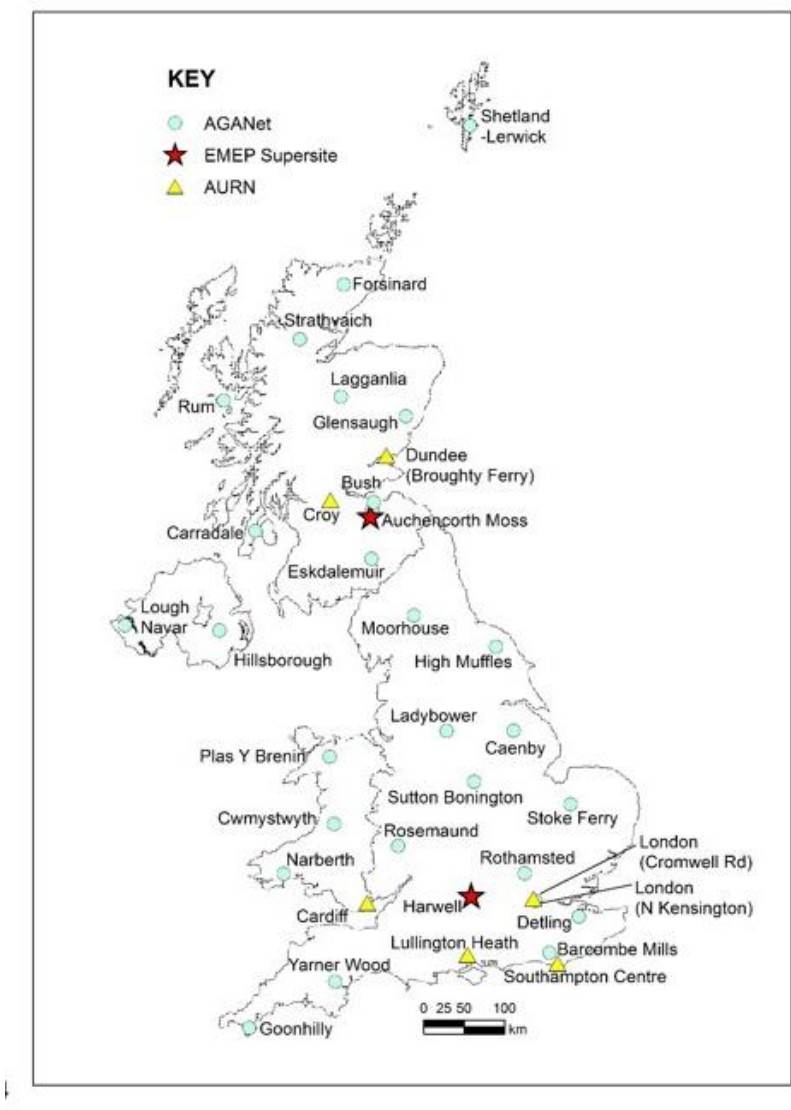

**Figure 1. Map of SO₂ monitoring sites in the UK used in this study. The AGANet sites provide monthly average concentrations, whilst the other sites report hourly values.**

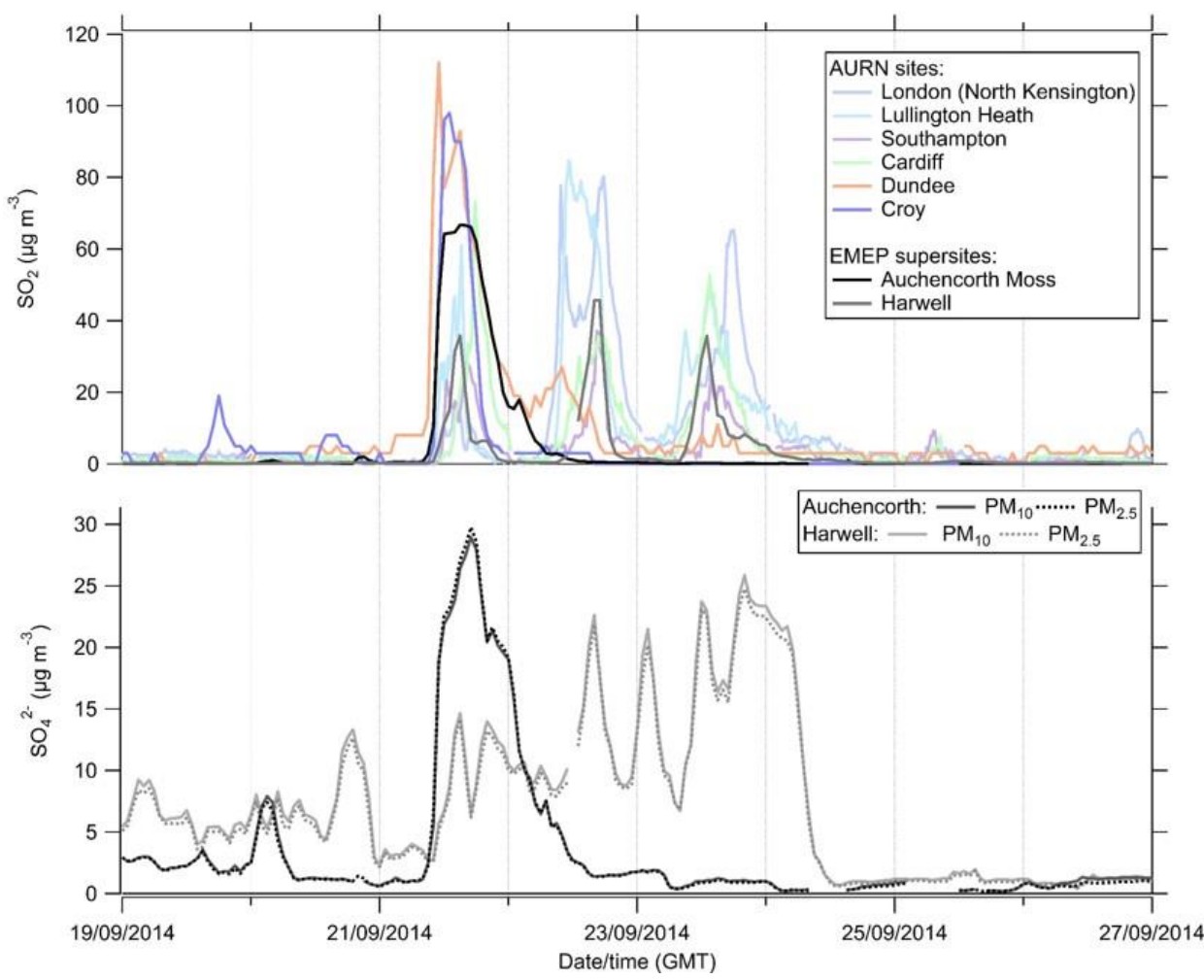

**Figure 2. Time series of SO₂ hourly measurements made at 6 AURN sites in the UK and the two UK EMEP supersites measurements of SO₂ and PM$_{10/2.5}$ SO$_4^{2-}$. (NOTE: SO₂ at Auchencorth Moss is underestimated between 11:00 and 22:00 (GMT) on the 21/09/14)**

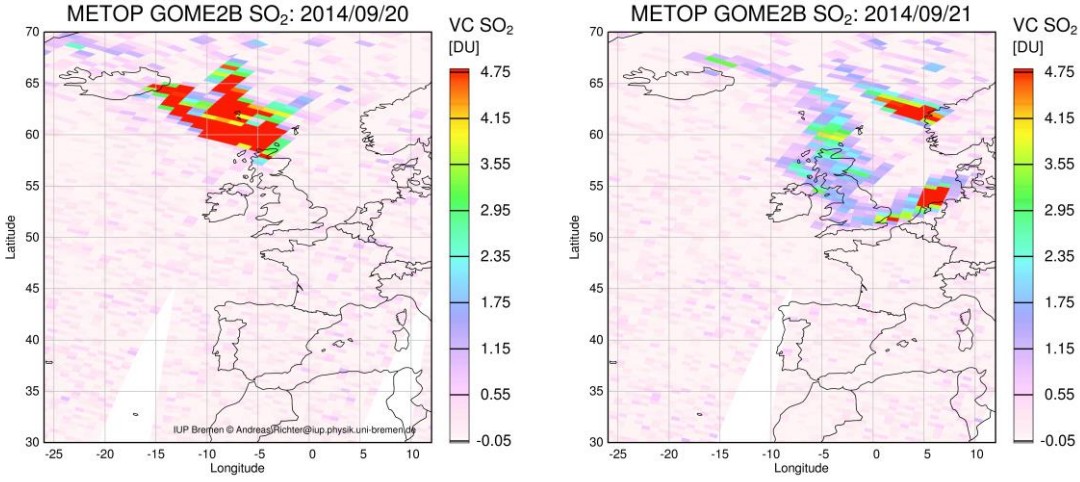

**Figure 3. Observation of the volcanic plume from Iceland to and across the UK by the GOME2B satellite instrument, taken at the satellite overpass around 9:30 local time. GOME2B measures SO₂ column density, where VC is the vertical column, which is the SO₂ concentration integrated vertically to provide a column density per unit surface area. SO₂ columns are given in Dobson Units (DU), the thickness the SO₂ layer would have at standard temperature and pressure in units of hundredths of a millimetre.**

s

# Daily SO$_2$ μgm$^{-3}$

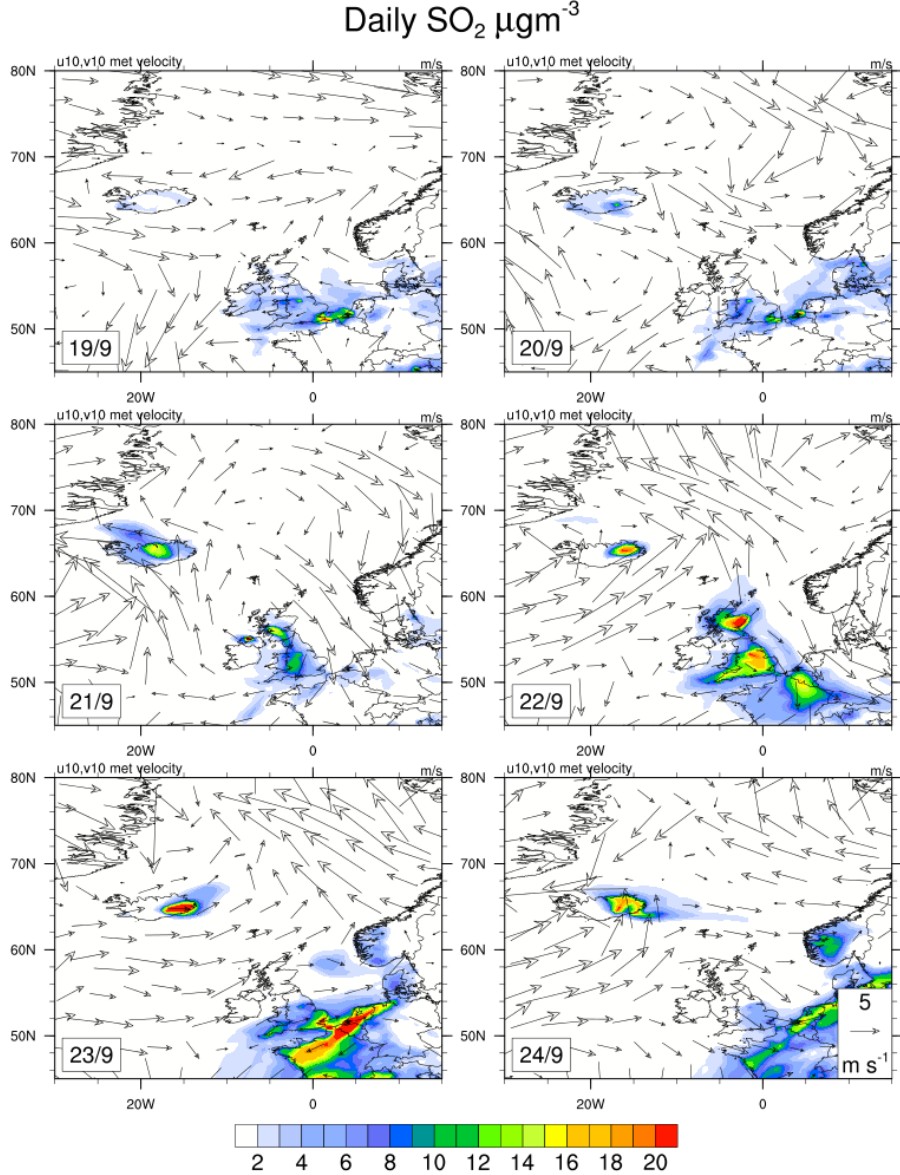

**Figure 4. Daily average surface concentration (μg m$^{-3}$) of the 19$^{th}$ - 24$^{th}$ of September 2014 of SO$_2$ calculated by the EMEP4UK model and the 12:00 of each days wind vector.**

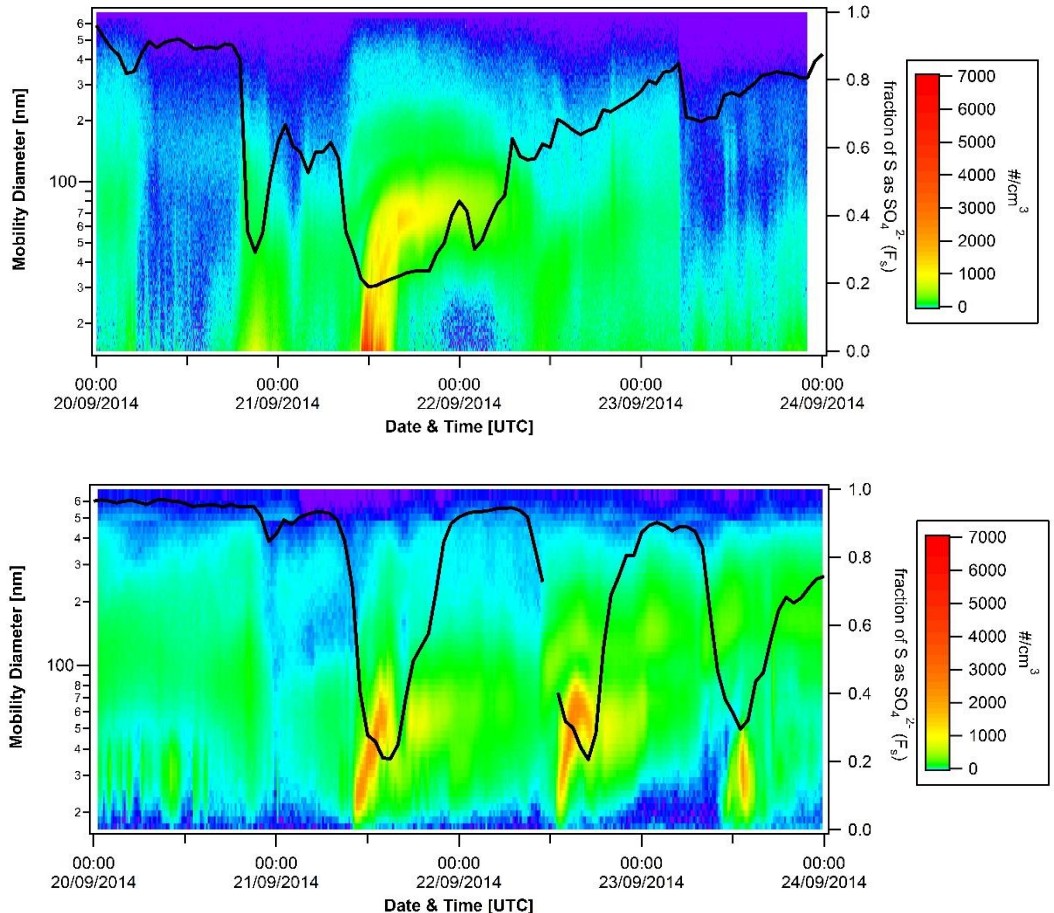

**Figure 5. Particle number concentration at Auchencorth Moss (top panel) and Harwell (bottom panel) (refer to Figure 1 for map) during the September 2014 volcanic plume event. Right hand y-axis is the Fs ratio (black line) measured by the MARGA for the same period, where lower Fs indicates 'younger' $SO_4^{2-}$. (Note: There are uncertainties regarding the size calibration of the instrument (see text), however the CPC was working correctly. The panel should therefore be regarded a qualitative indicator of an increase in the ultrafine particulate matter during the volcanic plume).**

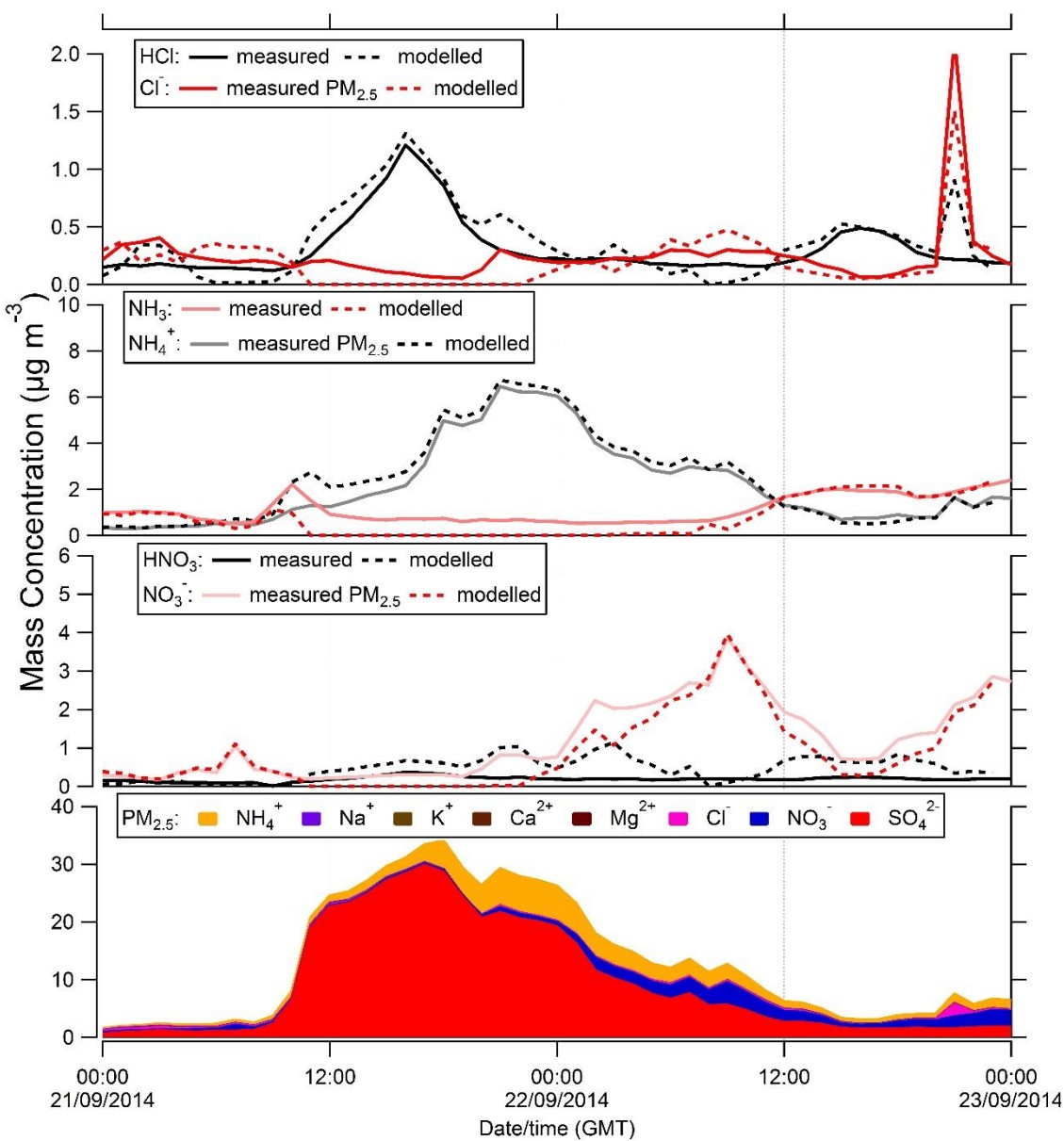

**Figure 6. Top three panels: thermodynamic partitioning of gas and aerosol modelled by ISORROPIA-II compared with the measured concentrations at Auchencorth Moss. The bottom panel shows a stacked representation of the chemical composition of PM₂.₅ at Auchencorth Moss as resolved by the MARGA instrument.**

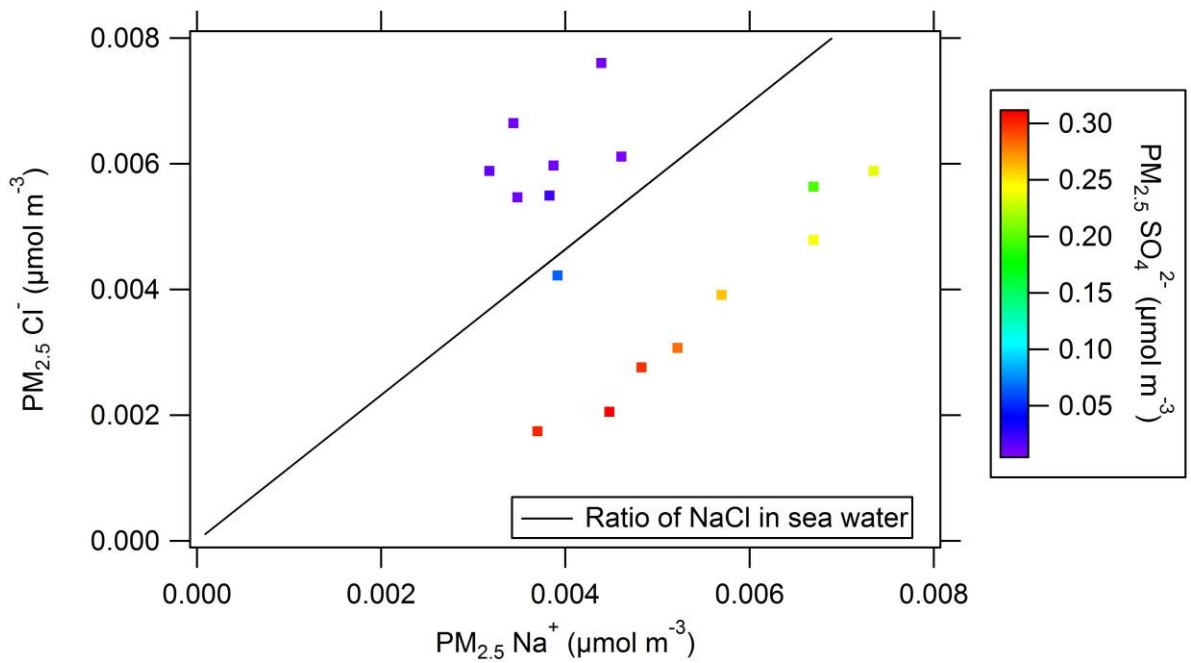

**Figure 7. Evidence of acid displacement in sea salt (PM₂.₅) on the 21st September 2014 from 00:00 to 18:00 (GMT) at Auchencorth Moss where the solid line is the known ratio of NaCl in sea water (Seinfeld and Pandis, 2006).**

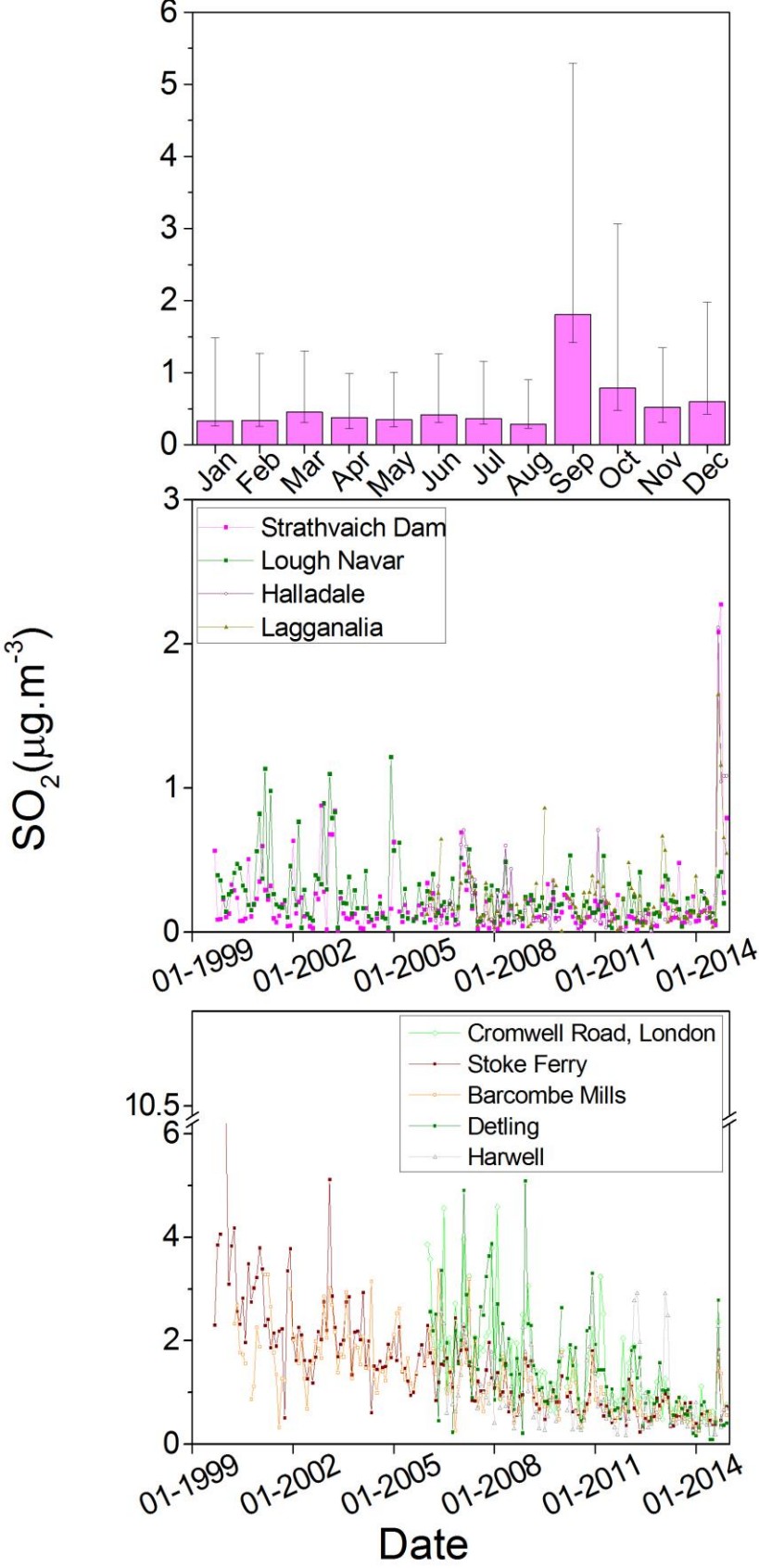

**Figure 8 UK Defra Acid Gas and Aerosol Network monthly SO₂. Top panel: 2014 monthly network average SO₂ concentration (30 sites, whiskers maximum and minimum values); Middle Panel: 5 remote sites in the network; Bottom Panel: 5 sites in southern England (Refer to Figure 1 map for location of sites).**

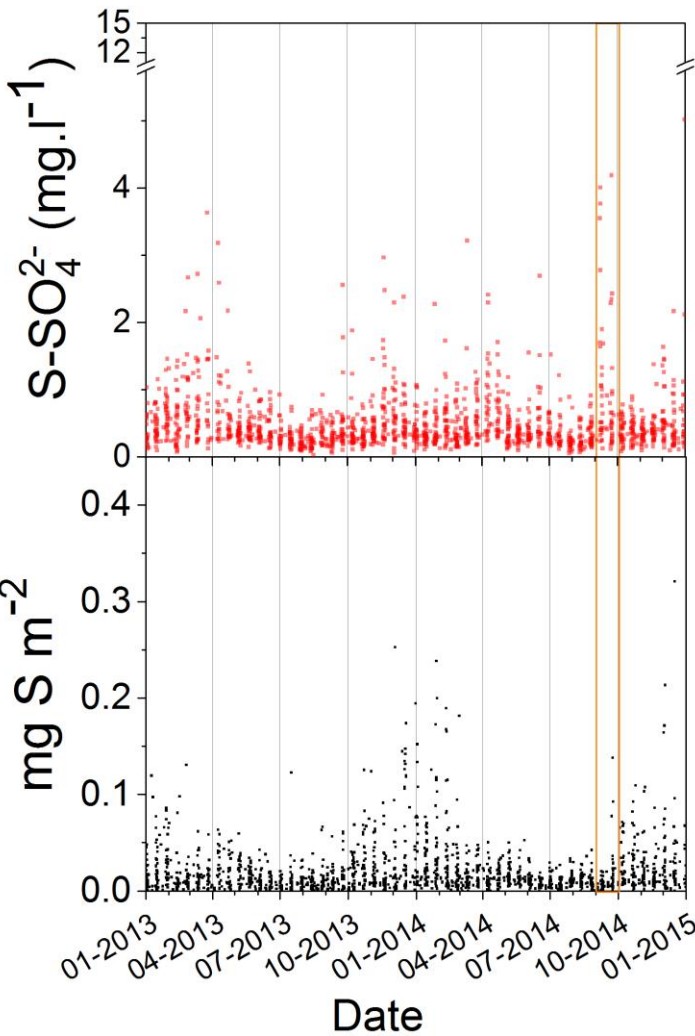

**Figure 9 UK Precip-Net data: All fortnightly site data for 2013 and 2014; Upper panel: S-SO$_4^{2-}$ concentrations; Lower panel: Sulfur deposition (mg S m$^{-2}$); Note fortnightly data with data plotted using the start date of the measurement period. (Data downloaded from UK-Air on 25/06/2015 and 02/02/2016). The orange box highlights the measurements in September 2014.**