# Peer review of "Impacts of the 2014-2015 Holuhraun eruption on the UK atmosphere"

_Atmospheric Chemistry and Physics, 2016_

## Referee Comment (RC1) · Anonymous Referee #1 · 20 Apr 2016

General:

This paper explores gas and particle phase composition and regional perturbations during an effusive volcanic event. The data presented are interested and certainly worth publishing; however, I have several major comments with respect to interpretation of the data that should be addressed before publication.

Major Comments:

The major comment I have is with respect to the simultaneous observation of newly formed particles and HCl. The authors imply that SO2 in the plume is responsible for both new particle formation and the heterogeneous displacement of chloride from sea

salt particles resulting in the formation of gaseous HCl. This finding is quite surprising considering that chloride displacement by H2SO4 should substantially deplete the concentrations of H2SO4 required to nucleate and grow new particles in the atmosphere. Figures 5 and 6 imply that heterogeneous uptake of H2SO4 and new particle formation by H2SO4 are occurring at the exact same time at the same site. Additional evidence is required to support this claim. Specifically, the authors should show that there is enough H2SO4 to support heterogeneous uptake, nucleation, and particle growth during this time period.

Specific Comments: Abstract: 1. The authors should clearly highlight that they are presenting data on effusive volcanic activity, which has been under-explored. This would further highlight the significance of their work.

2. Lines 25-27, only a few days in September are explored in depth. The authors should revise their statement that 4 months were studied and state that a large atmospheric perturbation occurring during a few days in Sept attributed to effusive volcanic activity is their main focus.

3. The authors should provide a sentence highlighting their lines of evidence that the perturbation was due to volcanic activity.

4. The authors should mention what the aerosol acidity was. This can be calculated using the data they have on hand and the ISORROPIA-II model.

Introduction

1. A more detailed discussion of explosive vs effusive volcanic activity would be helpful as well as a synopsis of previous finds relevant to tropospheric chemistry. This would help give context to the authors' findings.

2. The authors should comment on any findings relevant to halogen chemistry and volcanic activity if they are going to comment on HCl concentrations and their formation during a volcanic event.

Methods

1. If the SMPS had that much of a sizing offset, can any comment really be made regarding nucleation?

2. What temperature and RH conditions were used to run ISORROPIA-II?

Results

3.1 Identification of the volcanic plume

1. HYSPLIT back trajectories would help eliminate the possibility that other sources of aerosol are influencing the observations.

3.2.1 Formation of sulfate aerosols

2. The authors seem to imply that nucleation is occurring in the boundary layer at the same time as acid displacement. Further evidence is needed to support this. Can the authors prove that nucleation isn't occurring instead in the free troposphere where low particle surface area would favor this process?

3.2.2. Modification of the chemical composition within the plume

3. What does the temporal and spatial variability of aerosol Cl tell you? Do you see Na and Cl in proportions similar to sea salt near site that become more acidic with more SO4 in the aerosol as you move away from the plume?

4. Figure 7 is very hard to read and interpret. Why do none of the data fall on the 1:1 line?

---

## Referee Comment (RC2) · Anonymous Referee #2 · 5 May 2016

General Comments:

The paper is an important presentation of near-surface atmospheric composition measured during the passage of the Bardarbunga-Holuhraun volcanic gas cloud over the UK in 2014. This provides new data on the composition and aerosol size of such a cloud and its impact on other atmospheric constituents at very long range from the original eruption site. It raises interesting questions about the wider chemical influence of such a cloud, whilst also confirming that there was a low hazard to human health. This paper fits well with the existing literature on observations of this event, e.g. Schmidt, 2015; Gislason, 2015; Ialongo, 2015; Grahn, 2015, although it is noticeable that the latter two are not referenced.

The paper would benefit from some further discussion of the two main con-

cepts/theories it suggests and there are certain aspects that could be removed without detriment to the paper (see specific comments). There are also a relatively large number of typos that need correction.

Specific Comments:

The methods section of the report is very dry and is mainly a technical description of the equipment and sensors used. Depending on the editor's preference, there would be scope to move much of this to an appendix or even supplementary material. As it is, I would recommend that each section starts with a sentence summarising what it is that the technique measures. This is done for section 2.1 for example, but not for 2.3.

p2 Lines 15-17: The point the authors are trying to make here is a little unclear. Surely stratospheric measurements of volcanic composition are also only made/studied serendipitously? The fact that there is good satellite data of this eruption means that they cannot be referring to observations made with this type of sensor.

Section 2.2: The size range for Auchencorth Moss is provided, but not that for Harwell. It would be useful if the size range for Harwell could be included too.

Section 2.4 and 3.1: The authors need to provide information on where the GOME2 data has been obtained from and how it has been processed. Even though it is just being used as a qualitative picture, this information is still important. Further references for this would be useful in section 3.1. As the authors are using this data to compare to surface observations, the paper really needs some more information on the vertical sensitivity of the GOME2 instrument to SO2 and to make it clear to the reader that the satellite data in figure 3 shows SO2 throughout the atmosphere and not just the surface. SO2 in the satellite data does not necessarily correspond to increased SO2 at the surface.

Section 2.7: Given that two of the values generated by the statistical analysis are given in the abstract, I was expecting more substance to this aspect, but it appears

that it is only covered in section 2.7. A more detailed explanation of the approach is needed here, perhaps with an accompanying figure, to fully explain what has been done. For example, the period over which the AGANET data has been evaluated is not mentioned. In Table 1 it is not clear to me why the first column is average SO2 concentration. Section 2.3 would imply that the sensors record a monthly total – how has the average been calculated and is this particularly meaningful given that the authors nicely demonstrate elsewhere that the plume passed over in approx 4 days? It would be helpful for the authors to consider whether the concentrations from Sep 2014 are such outliers that actually the calculation of return period is not very robust? Some of these numbers are also referred to on page 8, where the authors refer to Goonhilly, but based on the numbers in Table 1 should this not be Yarner Wood instead? The phrasing on p8 also implies that these sites had the highest values of any site ever, whereas I think the authors are actually suggesting that these sites had their highest monthly values. I'd suggest making this clearer.

Section 3.1: Figure 2 (and Fig 6) nicely demonstrates that there were three "pulses" of SO2 observed across the UK from 21-25 Sep 2014 yet the authors make no mention of this. This observation in itself is interesting and the paper would benefit from consideration of this. It may not be possible to determine whether the cause is due to changing omissions at the source (a few days previous) or due to meteorological influences, but consideration of these aspects, and any others, should be included.

Section 3.1: The modelling sentence and Figure 4 are an unnecessary add-on and I recommend that the authors remove this together with section 2.5. The model does not add anything to the paper, but rather raises a whole number of questions that the authors will need to address if they want to include this. If anything, at first glance Figure 4 appears to contradict the other evidence as well as refute the descriptions in the text of northerly flow and that Scotland was worse affected than the south of the UK. There is not enough information on the modelling in the paper to explain what this figure is showing and describe why it does not match Figure 3. In addition, the fact that

the figure contour scale only goes to 22 ug/m3 but the observations in Fig 2 reach over 60-100 ug/m3 would suggest that the model is not doing a good job at representing the plume. Given the efforts by other authors (e.g. Schmidt et al 2015) to demonstrate that their models provide good agreement with satellite column loading data before applying them to near surface concentrations, the lack of any model validation in this paper is particularly stark.

Section 3.2: This is particularly interesting and one of the main new findings presented in this work. I have a few questions based on the text which the authors could hopefully easily incorporate the answers to. The fact that the difference in the plume aerosol diameter is so pronounced compared to the background is perhaps worth stating more clearly. P6 Line 32 – the text talks about the slow oxidation of SO2 in the troposphere, are you referring to heterogeneous or homogeneous oxidation here or both? P6 Line 37 – you refer to the plume containing "young SO4", I wonder whether it would be more appropriate to say that it is a "young plume"? Do you have any idea of the travel time since emission? Does this fit with your "young" finding? P7 Line 14-16: Is this true? This also sounds as though it needs consideration of the travel time. If the travel time was constant from the source to the observation point, then the particles arriving later in the day would have travelled longer in sunlight and so had a longer time to react. Perhaps it is just the wording of the sentence that needs tightening to make this clear. Should it be "with increasing time *after* sunrise"? And where you refer to "site" on line 16 do you mean the monitoring site or the eruption site?

Section 3.3: This section is also one of the main findings and theories in this paper derived from the observations. It is an interesting conclusion, but also raised a number of questions in my mind that it would be useful for the authors to comment on in the paper. Firstly, whether the displacement has occurred due to the transport of the plume over the sea for such a long distance (and/or time) or whether this is a relatively local affect due to the site being not far inland. Second, if the plume had travelled directly south, it actually would have been over land for many miles before reaching the site, how would

this fit the proposed mechanism? Third, does such a mechanism require transport to Scotland to have occurred near to the sea surface / within the boundary layer / or more local near-surface transport following above-BL transport over the ocean? Is there any data to support one of these over another?

Conclusions: The first line states that the eruption perturbed *all* aspects of the UK atmosphere. As a first point this should be the atmospheric composition (not atmosphere, see technical corrections), but even so this seems to be rather overstating what has been presented. For instance there is no mention in the paper of changes to oxidant levels, impact on ammonium reactions, etc, which is understandable given the context, but would be necessary to justify the "all aspects" claim. Some minor, but careful, rewording of this sentence would bring it more in line with what has actually been presented.

Technical Corrections:

p1 Line 25: "of the Holuhraun" needs modification to for example "of the Holuhraun fissure" or "at Holuhraun"

p1 Line 29: expand what EMEP stands for, or omit from sentence

p1 Line 34/35: missing "of" – "due to primary emissions of HCl"

p2 Line 4: delete one of the two "were"

p2 Line 8: add "of this type" to the end of the final sentence

p2 Line 22: add "is" to "but there is a very limited"

p2 Line 27 and p9 Line 30: the eruption was within the Bardarbunga volcanic system not the Holuhraun volcanic system. The eruption site and the eruption have been called Holuhraun.

P2 Line 29: recommend making "emission" plural, i.e. emissions

[Figure]

P2 Line 34: please explain what the EU-28 is/means for an international readership

P3 Line 8: "Northern" should have a lower case "n"

P3 line 26: replace "for" with "of", i.e. "a detailed description of the instrument"

P3 line 27: please expand the acronyms QA/QC

P3 line 27: replace "are" with "is", i.e. "by both instruments is given in"

P3 line 28: need to add "the", i.e. "between the Auchencorth"

P3 line 30: what is "IC"?

P3 line 30: need to add "a", i.e. "to achieve a lower detection"

P3 line 31: modify to be "therefore has an order of magnitude"

P4 line 26: a word is missing from "Downstream of is a gas"

P4 line 28: remove "this"

P4 line 35: Gome should be capitalised, i.e. GOME

P5 line 16: remove "below"

P5 line 24: in a number of places in the text the authors use "high resolution analysis", this is not specific enough, I assume that they mean high temporal resolution not spatial? This should be included/made clear.

P6 line 19: there are other references that could be included here for the observation of the plume (see General Comments). It would also be useful for the authors to clarify whether these observations occurred at approximately the same time (i.e. related to the same plume transport) or at different times during the prolonged eruption.

P7 line 4: change to "or 'banana' shape in Figure 5, starting with"

P7 line 23-25: change line 23 to be "air quality impact from particulates during" and

remove "due to particles" from line 25.

P7 line 35-36: The reference list is not needed here as these are already referred to or implied earlier in this sentence.

P7 line 38: However, a reference is definitely needed for the molar ratio of HCl/SO2 being <1% near source.

P8 line 4: Is HCl correct at the end of this line? Should it be Cl-?

P8 line 31: Add "at", i.e. "in particular at the sites", and South West should be lower case

P8 line 32: add "in", i.e. "whereas in Northern Ireland", and change "were" to "was"

P8 line 33: add "that", i.e,. "noted, however, that there"

P8 line 37: suggest rephrasing "was not important to" to "was not significantly different to normal" or similar

P9 line 4: change to "the UK atmospheric composition during the latter part..."

P9 line 5: change line to "Elevated SO2 was observed by the networks at both high and low temporal resolution. These observations complement the study by"

P9 line 8: remove the comma and change "to" to "in"

P9 line 10-11: I think we would expect particle formation and growth to be occurring in the plume based on past chemical and physical knowledge, so it would be better to say "... from the two EMEP supersites provide observational evidence for new particle formation and growth occurring as the plume..."

P9 line 14: add "work", i.e. "the recent modelling work undertaken"

P9 line 22: add "that" to become "The study has highlighted that even though"

P9 line 25: change "are" to "is"

P9 Line 26: remove "the" from "concurrently with the SO2"

Fig 1 caption: Repeat of "sites"

Fig 3 caption: explain what VC SO2 is and what DU is. Are these images snap-shots or aggregated daily totals or means? This needs to be stated.

Fig 5 caption: explain what the black line is

Fig 7 caption: remove capitalisation from "Sea". It would be useful to explain the colouration of the dots in the main paper text and what this means for this event.

Fig 9 caption: what are the orange lines?

[Figure]

---

## Author Comment (AC1) · 12 Aug 2016

Responses to Reviewers

Manuscript acp-2016-177

5   **Impacts of the 2014–2015 Holuhraun eruption on the UK atmosphere**

**by Marsailidh M. Twigg *et al.***

10  The authors would like to thank the reviewers for the time spent reviewing the manuscript. Please find below in black the reviewer comments and in BLUE the authors' response to each of the comments. We have revised the manuscript to implement the recommendations from the reviewers, which can be found at the end of this document using track changes.

**Anonymous Referee #1 ()

*General:*
This paper explores gas and particle phase composition and regional perturbations
20  during an effusive volcanic event. The data presented are interested and certainly worth publishing; however, I have several major comments with respect to interpretation of the data that should be addressed before publication.
**RESPONSE:** We are pleased that the reviewer believes the data to be of interest and worth publishing. We have addressed all comments raised by the reviewer and
25  hope that the major concerns that reviewer had about the manuscript have been dealt satisfactorily.

*Major Comments:*
The major comment I have is with respect to the simultaneous observation of newly
30  formed particles and HCl. The authors imply that $SO_2$ in the plume is responsible for both new particle formation and the heterogeneous displacement of chloride from sea salt particles resulting in the formation of gaseous HCl. This finding is quite surprising considering that chloride displacement by $H_2SO_4$ should substantially deplete the concentrations of $H_2SO_4$ required to nucleate and grow new particles in
35  the atmosphere. Figures 5 and 6 imply that heterogeneous uptake of $H_2SO_4$ and new particle formation by $H_2SO_4$ are occurring at the exact same time at the same site. Additional evidence is required to support this claim.

Specifically, the authors should show that there is enough $H_2SO_4$ to support
40  heterogeneous uptake, nucleation, and particle growth during this time period.

**RESPONSE:**
The information on the full plume composition at source is too incomplete for detailed plume modelling that would allow the prediction of OH and $H_2SO_4$ concentrations
45  during the plume transport. We agree that it is somewhat surprising that nucleation and acid displacement should occur in parallel as one would expect one process to be favoured over the other, depending on the aerosol surface area.

Instead, our starting point are the observations. There is clear evidence for particle nucleation which is hardly ever observed at Auchencorth. There is also clear evidence for HCl displacement from sea salt. This is evident in the high HCl concentrations, and in the fact that the $Cl^-/Na^+$ ratio is lower than that of sea water (Fig. 7). Although the latter could also be due to displacement by $HNO_3$, there are no major sources of NOx along the trajectory and $HNO_3$ concentrations are significantly lower (especially in terms of mixing ratio) than those of HCl (Fig. 6).

We can only speculate why heterogeneous chemistry and nucleation may take place simultaneously. The fact that they are observed at the same time does not necessarily mean that they occur at the same time and at the same location, with the timing of observation determined by when the plume reached the sampling point. For the displacement reaction to occur, the sea spray aerosol needs to mix with the volcanic plume from aloft. It is feasible that this mixing generates 'pockets of air' that are dominated by the volcanically influenced air (where nucleation is favoured) and others which have efficiently mixed with the sea salt (where heterogeneous chemistry is favoured). Through continued mixing and through the long-term integration of the measurement both would be reported at the same time. Similarly, the conditions that favour one process over the other may vary along the trajectory between Iceland and the UK.

We also cannot rule out that other compounds are taking part in the nucleation process and aerosol growth and have added this caveat to the manuscript:

*"In addition, we cannot rule out that other compounds are taking part in the nucleation process and aerosol growth."*

We have also included the above discussion into the manuscript text at the end of section 3.2.2:

*"It is surprising that both acid displacement and nucleation events would be observed in parallel, as one process would be expected to be favoured over the other, depending on the aerosol surface area. Instead it is hypothesized that though both events are observed at the same time, it does not necessarily mean that they occur at the same time and at the same location. For the displacement reaction to occur, the sea spray aerosol needs to mix with the volcanic plume from aloft. It is feasible that this mixing generates 'pockets of air' that are dominated by the volcanically influenced air (where nucleation is favoured) and others which have efficiently mixed with the sea salt (where heterogeneous chemistry is favoured). Through continued mixing and through the long-term integration of the measurement both would be reported at the same time. Similarly, the conditions that favour one process over the other may vary along the trajectory between Iceland and the UK."*

*Specific Comments:*
Abstract: 1. The authors should clearly highlight that they are presenting data on effusive volcanic activity, which has been under-explored. This would further highlight the significance of their work.

**RESPONSE:** The abstract has been modified to try and highlight the effusive eruption aspect and the paucity of data from this type of eruption further:

Page 2 lines 13-14
*"The measurements can be used to both challenge and verify existing atmospheric chemistry of volcano plumes, especially those originating from effusive eruptions, which have been under-explored, due to limited observations available in the literature."*

2. Lines 25-27, only a few days in September are explored in depth. The authors should revise their statement that 4 months were studied and state that a large atmospheric perturbation occurring during a few days in Sept attributed to effusive volcanic activity is their main focus.
**RESPONSE:** We agree the main focus of data analysis was based on a few days during a specific volcanic plume. The abstract now states:

*"This study focuses one major incursion in September 2014, affecting the surface concentrations of both aerosols and gases across the UK, with sites in Scotland experiencing the highest sulfur dioxide ($SO_2$) concentrations."*

Furthermore to highlight the 4 months of low resolution network data additional text has been added to the abstract to highlight this.

*"Volcano plume episodes were periodically observed by the majority of the UK air quality monitoring networks during the first 4 months of the eruption (August – December 2014), at both hourly and monthly resolution. In the low resolution networks, which provides monthly $SO_2$ averages, concentrations were found to be significantly elevated at remote "clean" sites in NE Scotland and SW England, with record high $SO_2$ concentrations for some sites in September 2014."*

In addition we have modified our introduction text too:

*"This paper studies the volcanic impact on the UK atmosphere and focuses on one major incursion in September 2014 during the Holuhraun eruption and provides the first evidence of wide scale effects, based on the measurements from the UK air quality monitoring networks which deliver data at both high (hourly) and low (monthly) temporal resolution."*

3. The authors should provide a sentence highlighting their lines of evidence that the perturbation was due to volcanic activity.
**RESPONSE:** Additional text has been added to the abstract to state how we confirmed that the perturbation was due to volcanic activity.

Page 1 lines 28-30
*"The perturbation event observed was confirmed to originate from the fissure eruption by using satellite data from GOME2B and the chemical transport model, EMEP4UK, which was used to establish the spatial distribution of the plume over the UK."*

4. The authors should mention what the aerosol acidity was. This can be calculated using the data they have on hand and the ISORROPIA-II model.
**RESPONSE:** We have now calculated the pH for the period using the ISORROPIA model and placed it in the discussion and commented on it in the text of the abstract too. The edits are as follows:

*P2 Line 1-2:*
*"It was confirmed using the chemical thermodynamic model, ISORROPIA-II, that aerosol measured was acidic with an estimated pH of 3.80 during the peak of the event."*

Section 3.2.2:
*"The aerosol at the Auchencorth was found to only become acidic during the period when the plume was evident, using the results of $[H^+]$ calculated from ISORROPIA-II model to calculate pH. During the plume event, the pH dropped from pH 7.97 at 09:00 GMT to pH 3.80 at 15:00 on the 21/09/14. This is unusual for the site as it has been previously noted that aerosols at site are generally basic in nature, due to an excess of $NH_4^+$ aerosol, when ion balance studies have been carried out (Twigg et al., 2015)."*

Introduction
1. A more detailed discussion of explosive vs effusive volcanic activity would be helpful as well as a synopsis of previous finds relevant to tropospheric chemistry. This would help give context to the authors' findings.
**RESPONSE:** Though the subject was covered an expansion of this section has been written providing additional text to the introduction to highlight 1) diffusive vs explosive eruptions and 2) a synopsis of previous studies with regards to current knowledge of tropospheric chemistry within a distal volcanic plume has also been added. Overall the introduction has been expanded to cover the subject area more fully. The revised text is below:

[revised manuscript text omitted]
 authors should comment on any findings relevant to halogen chemistry and volcanic activity if they are going to comment on HCl concentrations and their formation during a volcanic event.
**RESPONSE:** We have expanded both the introduction (refer to point 1 above) on halogen chemistry. In addition we have highlight in section 3.2.2 of the discussion the importance of halogen chemistry and the next steps required in order to comment further on the observed chloride chemistry within this volcanic plume.

*"Roberts et al. (2009) has previously modelled the likely perturbations within a volcano plume upon mixing with background air, which found that both ozone and $NO_x$ were perturbed with bromine chemistry as the key driver. In order understand the net impact of this eruption on the chemical perturbation of background sea salt aerosol and the production of HCl in the gas phase, there would need to be a full*

*chemical transport model which includes both chloride and bromine chemistry, which is beyond the scope of this paper."*

Methods
1. If the SMPS had that much of a sizing offset, can any comment really be made regarding nucleation?
**RESPONSE:**
We agree that due to the offset that we cannot provide direct evidence of nucleation, however the shape of the size distribution would suggest that nucleation had occurred within the volcanic plume (please refer above to major concerns response for further details). In the methods with regards to the offset at Auchencorth Moss, we clearly state the SMPS is a qualitative indicator of increase in ultrafine particles but the size distribution could not be verified.

2. What temperature and RH conditions were used to run ISORROPIA-II?
**RESPONSE:**
We used the measured temperature and RH on the ISORROPIA-II run. We have added additional text to state this:

*"The model was run using as an input the bulk (i.e. gas + aerosol) concentration of all compounds (ammonium, nitrate, sulfate and chloride) measured by the MARGA (input in µmol m$^{-3}$) with measured temperature and relative humidity and operated in the metastable, forward reaction."*

Results
3.1 Identification of the volcanic plume 1. HYSPLIT back trajectories would help eliminate the possibility that other sources of aerosol are influencing the observations.
**RESPONSE:**
We chose not to use the HYSPLIT model as the EMEP4UK model presented is able to do the same by illustrating the distribution and the direction of the plume. We have added to supplementary material the plume run using the HYSPLIT model for the peak days at both sites, please refer to figures S1 and S2. In addition to this we have noted the additional evidence in the main manuscript where it states:

*"Modelling of the plume by EMEP4UK further confirmed the volcanic origin and dispersion of the observed SO$_2$ plumes both at Harwell and Auchencorth Moss (Figure 4), as did back trajectories using the HYSPLIT model, which can be found in figures S1-S2. To provide evidence that the EMEP4UK model was able to replicate the spatial distribution of the plume a comparison of the time series of observations to the model at both sites is given the supplementary material (figure S3)."*

3.2.1 Formation of sulfate aerosols
2. The authors seem to imply that nucleation is occurring in the boundary layer at the same time as acid displacement. Further evidence is needed to support this. Can the authors prove that nucleation isn't occurring instead in the free troposphere where low particle surface area would favor this process?

**RESPONSE:** Please refer to our response under major concerns with regards to nucleation and acid displacement being observed co-currently. In addition, there is evidence, which we have included in the text, showing nucleation has been observed in polluted environments. Recent literature has confirmed that nucleation can take place in the presence of background aerosols, even within ash rich volcanic plumes (Boulon et al., 2011), however this study was in the free troposphere. There are also examples of nucleation and particle growth in polluted environments measured at the surface, including Hamed et al. (2007) who observed nucleation in the Po Valley at the site San Pietro Capofiume. Hamed et al. (2007) also cites other literature of nucleation studies within polluted areas including Kulmala et al., 2005 in New Delhi, India and Alam et al. (2003) in the Birmingham, UK. Both of these studies observed nucleation at the surface. This additional information has been added to back up the text that there is already examples in the literature of surface observations of nucleation,

**Section 3.2.1:**
**"***Though nucleation is thought to occur in the free troposphere where there is a low particle surface area there is increasing evidence in the literature of nucleation events measured at the surface in polluted environments, such as Hamed et al. (2007) who observed nucleation events in the Po Valley, Italy, and by Kulmala et al. (2005) in New Delhi.*"

3.2.2. Modification of the chemical composition within the plume
3. What does the temporal and spatial variability of aerosol Cl tell you? Do you see Na and Cl in proportions similar to sea salt near site that become more acidic with more SO4 in the aerosol as you move away from the plume?
**RESPONSE:**

We cannot comment on the spatial distribution across the UK for sea salt, as the period presented, $Na^+$ and $Cl^-$ were frequently close to the detection limit at Harwell. We should have highlighted that there is prior knowledge with regards to sea salt and acid displacement events at Auchencorth Moss. Twigg *et al* (2015) has demonstrated that on some occasions where high $NO_3^-$ events have been observed to coincide, on occasions, in a depletion of $Cl^-$, which is thought to occur through the interaction of $HNO_3$ with sea salt. It is assumed that this observation is due to the location of the site, which is close to the sea and has been found to abundant in sea salt (35% in annual average of water soluble inorganic $PM_{2.5}$ ( Twigg *et al.* (2015)) . We have added the following text to try and reflect this:

*"Further evidence of acid displacement was found at Auchencorth Moss when the ratio of $Na_+$ and $Cl^-$ was compared to the known ratio of sea water, where a large relative depletion of aerosol $Cl^-$ was found during elevated $SO_4^{2-}$, represented by a change in colour of the markers in Figure 7. This is not the first time the site has observed acid displacement, Twigg et al. (2015) observed the site to be rich in sea salt due to its proximity to the sea, with 35% of the annual average of the inorganic composition of $PM_{2.5}$ attributed to sea salt. During high nitrate ($NO_3^-$) episodes it was observed on occasions that this coincided with an apparent depletion of $Cl^-$ from sea salt, which was attributed to the displacement of $Cl^-$ by $HNO_3$. "*

4. Figure 7 is very hard to read and interpret. Why do none of the data fall on the 1:1 line?

**RESPONSE:** The line presented is not a 1:1 line but the known ratio of Na:Cl in seawater, which was stated both in the legend and in the figure text, citing the reference for the calculation. An excess of Cl⁻ is possible, as there are other Cl⁻ sources, whereas Na⁺ is generally thought to originate primarily from sea spray. We have edited the fonts and markers in figure 7, as well as updated the figure text to make the message clearer.

**Anonymous Referee #2** ()

General Comments:

The paper is an important presentation of near-surface atmospheric composition measured during the passage of the Bardarbunga-Holuhraun volcanic gas cloud over the UK in 2014. This provides new data on the composition and aerosol size of such a cloud and its impact on other atmospheric constituents at very long range from the original eruption site. It raises interesting questions about the wider chemical influence of such a cloud, whilst also confirming that there was a low hazard to human health. This paper fits well with the existing literature on observations of this event, e.g. Schmidt, 2015; Gislason, 2015; Ialongo, 2015; Grahn, 2015, although it is noticeable that the latter two are not referenced. The paper would benefit from some further discussion of the two main concepts/theories it suggests and there are certain aspects that could be removed without detriment to the paper (see specific comments). There are also a relatively large number of typos that need correction.

**RESPONSE:** The authors are glad that the reviewer thought this paper was an important presentation of the near-surface atmospheric composition following the Bardarbunga-Holuhraun volcanic gas cloud in 2014 and the reviewer for bringing the other new references to our attention. We have modified our introduction to ensure that Ialongo *et al.*, 2015 and Grahn *et al.*, 2015 have also been included.

*"The ground level concentration of $SO_2$ exceeded the hourly health limit (350 µg m$^{-3}$) over much of the country for periods of up to several weeks (Gíslason et al., 2015) and there were complaints as far as Scandinavia of a foul smell, which has been attributed to sulfurous compounds from the fissure eruption using satellite data (Grahn et al., 2015). Exceedances in the hourly health limits were also observed periodically in Northern Finland at surface observation sites, which were confirmed by satellite observations (Ialongo et al., 2015)."*

Specific Comments:

The methods section of the report is very dry and is mainly a technical description of the equipment and sensors used. Depending on the editor's preference, there would be scope to move much of this to an appendix or even supplementary material. As it is, I would recommend that each section starts with a sentence summarising what it is that the technique measures. This is done for section 2.1 for example, but not for 2.3.

**RESPONSE:** We agree that the method section is technical but needs to be well described to demonstrate both QA/QC in the methods. We have consulted the editor who has agreed that we should keep the methods within the main text of the manuscript. We however have added a short summary to start of each section in the

methods to describe what each technique measures, in order that reader can decide if they wish to read the technical description. Please refer to modified manuscript for additions.

5    p2 Lines 15-17: The point the authors are trying to make here is a little unclear. Surely stratospheric measurements of volcanic composition are also only made/studied serendipitously? The fact that there is good satellite data of this eruption means that they cannot be referring to observations made with this type of sensor.

**RESPONSE:** The reviewer is correct that all volcano measurements are serendipitous in that if you have the observation system in place you can observe the plume and the sentence was not clearly focused. Stratospheric and tropospheric satellite characterisation of plumes is now global and much more information can be
15    derived from them than in the past. The serendipity refers to the fact that instances when the background conditions with the volcano chemistry and physics are monitored in real time can only happen when a plume passes a highly instrumented location e.g. air quality stations. Due to this comment and reviewer 1s comments, the introduction has been revised and expanded to discuss in more detail. Specifically,
20    the serendipity sentence has been replaced by:

*"The impact of effusive eruptions on the troposphere at both local and regional scales, are most frequently studied in responsive mode, post eruption initiation. As technology and instrumentation has developed, global air quality monitoring effort*
25    *has increased, resulting in some cases where background conditions and the evolution of the plumes can be studied."*

Section 2.2: The size range for Auchencorth Moss is provided, but not that for Harwell. It would be useful if the size range for Harwell could be included too.

**RESPONSE:** We apologise for missing out the size ranges from Harwell. They have now been included in the text.
 *"At Harwell, aerosol number size distributions measured in the range of 16.55 to 604.3 nm by the SMPS (Electrostatic classifier 3080, differential mobility analyser*
35    *3081, and condensation particle counter 3775, all TSI Inc.)."*

Section 2.4: The authors need to provide information on where the GOME2 data has been obtained from and how it has been processed. Even though it is just being used as a qualitative picture, this information is still important.
40    **RESPONSE:** The GOME2 data product used is the volcanic alert product from the University of Bremen. To provide more information to the reader, the following paragraph has been added:

*"Satellite UV/vis retrievals yield integrated vertical columns of absorbing species and*
45    *usually do not provide information about the vertical distribution of a trace gas. As the sensitivity of the observations decreases towards the surface, an assumption has to be made in the retrieval on the vertical profile of the target species in order to apply appropriate weights called air mass factors. Here, the standard volcanic product from*

*the University of Bremen is used (http://www.iup.uni-bremen.de/doas/gome2_so2_alert.htm) which assumes a volcanic eruption profile with an $SO_2$ peak at 10 km height."*

Section 3.1: Further references for this would be useful in section 3.1. As the authors are using this data to compare to surface observations, the paper really needs some more information on the vertical sensitivity of the GOME2 instrument to $SO_2$ and to make it clear to the reader that the satellite data in figure 3 shows $SO_2$ throughout the atmosphere and not just the surface. $SO_2$ in the satellite data does not necessarily correspond to increased $SO_2$ at the surface.

**RESPONSE:** We agree that surface observations and satellite column data do not measure the same quantity and therefore not necessarily detect $SO_2$ from the same source. However, we think the coincidence of the two measurements provides strong indication that in this case, volcanic $SO_2$ is detected at the surface. In response to the reviewer's comment, we have expanded the discussion with the following paragraph:

*"While satellite observations only provide tropospheric columns of $SO_2$ and not the concentrations at surface which are reported by the ground-based measurements, they do provide important information on the origin of enhanced $SO_2$ values. As the sensitivity of satellite retrievals to surface $SO_2$ is low, sulfur dioxide from surface pollution can rarely be detected over Europe. Volcanic plumes in contrast can readily be observed in the data as they usually extend to higher altitudes and contain much larger vertical columns of $SO_2$. The fact that several Dobson Units of $SO_2$ are observed in GOME2 data on September 20th and 21st, 2014 over the UK, and that the $SO_2$ plume originates from Iceland strongly indicates that the observed $SO_2$ is of volcanic origin. It is therefore very probably that simultaneous $SO_2$ enhancements measured at the surface are also linked to the volcanic emissions."*

Section 2.7: Given that two of the values generated by the statistical analysis are given in the abstract, I was expecting more substance to this aspect, but it appears that it is only covered in section 2.7. A more detailed explanation of the approach is needed here, perhaps with an accompanying figure, to fully explain what has been done. For example, the period over which the AGANET data has been evaluated is not mentioned.

**RESPONSE:** The reviewer is correct that this part of the work is only briefly described with the methodology, partly due to the range of methods covered in this paper. The method is covered in section 2.7 and the outcomes in the results section 3.3. To address this lack of detail we have expanded to include a bit more detail in both sections and added references to the statistical method. We have not added an additional figure but have referred to Figure 8 when discussing underlying trends. Thank you to the reviewer for spotting that we did not highlight the operational dates for AGANet, it was an omission which we missed. These have been added into the text. Revised text is as follows:

*"As well as high temporal resolution analysis (hourly measurements) of the volcanic plume, the monthly average $SO_2$ from AGANet measurements (Section 2.3) were analysed to assess the impact of the fissure eruption on the background atmosphere across the UK. The plume was clearly observable in this dataset. In order to understand the relative magnitude of the perturbation on a national scale for both air*

*quality and sulfur deposition, against a background of decreasing anthropogenic emissions a statistical treatment was thought to be appropriate. Specifically the likelihood of a reoccurrence of the observed concentrations in the UK background was calculated as many AGANet sites show decreasing trends over time for $SO_2$ and $SO_4^{2-}$, reflecting decreases in emissions. This is observed for both the annual mean concentration and the annual maximum concentration. A high concentration superimposed on a downward concentration trend in a measurement series over a period appears to be a less unlikely observation at the end of the time period than at the beginning. Therefore the AGANet measurement data were adjusted to remove any underlying trends before further analysis to assess the unusualness of the September 2014 elevated $SO_2$ concentrations. Theoretically exceedances over a threshold follow a Pareto distribution and the threshold was chosen by fitting an 85% quartile regression to this dataset using a smoothing spline for each site individually (i.e. there was no assumption of a general trend). This methodology has previously been developed much further for application to case studies with substantially more data (Chavez-Demoulin and Davison, 2005;Northrop and Jonathan, 2011;Reich et al., 2011). The fitted Pareto distribution generates the probabilities of occurrence of the elevated concentrations – which in this case were associated with the volcanic plume. This is expressed in Table 1 as a return probability and return time, which is the statistical likelihood of a similar concentration to be observed again based on the long term trend of $SO_2$ over the 1999-2014 period at each site expressed in the resolution of the measurements."*

**A short section has been added to the results, section 3.3:**

*"At sites which are much more anthropogenically influenced, though the plume is clearly observable, the $SO_2$ concentration is unremarkable with return probabilities of >$1x10^{-1}$ (e.g. Detling). A few sites on the Western side of the UK were not in the pathway of the plume therefore no elevated concentrations were observed, e.g. Rum."*

In Table 1 it is not clear to me why the first column is average $SO_2$ concentration. Section 2.3 would imply that the sensors record a monthly total – how has the average been calculated and is this particularly meaningful given that the authors nicely demonstrate elsewhere that the plume passed over in approx 4 days?
 **RESPONSE:** The DELTA samplers pass a known air flow through a coated denuder for a one month exposure period, therefore the analytical results are the total amount of sulfate collected in the volume of sampled air, hence the average concentration over the month is recorded. AGANet and the Precip-Net are designed to underpin the UKs national concentration and deposition mapping, feeding into the critical level and load assessments under UNECE. The assessments use annual average concentrations. The monthly average data is meaningful as much as hourly average is of the variability within it or an annual average temperature is. Given the evidence from the monthly average measurement network shows the extent of the plume across the UK in the context of the national chemical climate we would contend that it is meaningful information.

It would be helpful for the authors to consider whether the concentrations from Sep 2014 are such outliers that actually the calculation of return period is not very robust?

**RESPONSE:** Return period is one way of measuring how extreme (or how much an outlier) the concentration is. Hence a large return time indicates a very low probability of the concentration occurring again given the current underlying trend in the dataset, i.e. the concentration is extremely unusual. We have edited the text to explain this more clearly.

Some of these numbers are also referred to on page 8, where the authors refer to Goonhilly, but based on the numbers in Table 1 should this not be Yarner Wood instead? The phrasing on p8 also implies that these sites had the highest values of any site ever, whereas I think the authors are actually suggesting that these sites had their highest monthly values. I'd suggest making this clearer.

**RESPONSE:** We apologise this should have read Yarner Wood and not Goonhilly. We have corrected the manuscript text. We have also edited the text as the reviewer is correct that the site had its highest monthly value.

*"Similarly Yarner Wood in the south west of England experienced the highest concentrations on record for the site, and even taking into account the underlying decreasing trend in $SO_2$ concentrations, return probabilities were as low as $3 \times 10^{-4}$ (Table 1, refer to section 2.7 for statistical methods)."*

Section 3.1: Figure 2 (and Fig 6) nicely demonstrates that there were three "pulses" of $SO_2$ observed across the UK from 21-25 Sep 2014 yet the authors make no mention of this. This observation in itself is interesting and the paper would benefit from consideration of this. It may not be possible to determine whether the cause is due to changing omissions at the source (a few days previous) or due to meteorological influences, but consideration of these aspects, and any others, should be included.

**RESPONSE:** We agree we did not highlight the 3 pulses observed at the Harwell site and have reworded the text accordingly. The use of the EMEP4UK model was able to illustrate that the pulses were due to meteorological conditions, however variables such as injection height of the plume at source, frequently changing emission rates at source and the oxidation rates to form $SO_4^{2-}$ can explain the variation between the modelled and measured plumes. We have reviewed the EMEP4UK model injection height and edited the methods and Fig 4 accordingly, however a full model vs observation study is beyond the scope of this work. We have added additional text and figures to the supplementary material to confirm this.

**Methods:**
*"The emissions are injected into the model vertical column equally from the 1km to 3 km."*

**Supplementary information:**

*"The EMEP4UK model was also used to confirm the distribution of the plume, as presented in Figure 4 in the main text. To provide further evidence of the agreement of the spatial distribution of the plume by the model, data for the sites Auchencorth Moss and Harwell data were plotted in a time series against the observed*

*concentrations. It is clear to see that the model is able to show that Auchencorth Moss the observed plume on the 21/09/14, however the site did not observe the plume predicated on the 22/09/14 or match the magnitude of the plume at the surface. At Harwell, the observed 3 pulses on consecutive days (21/09/14- 23/09/14) from the volcanic plume were identified in the temporal pattern but again the magnitude of the surface concentration is underestimated by EMEP4UK. The number of explanations why the magnitude and even the spatial of the distribution of the plume was not comparable to surface concentration measurements."*

[Figure]

**Figures S3 Times series of modelled plume by EMEP4UK and that observed by the MARGA instrument."**

Section 3.1: The modelling sentence and Figure 4 are an unnecessary add-on and I recommend that the authors remove this together with section 2.5. The model does not add anything to the paper, but rather raises a whole number of questions that the authors will need to address if they want to include this. If anything, at first glance Figure 4 appears to contradict the other evidence as well as refute the descriptions in the text of northerly flow and that Scotland was worse affected than the south of the UK. There is not enough information on the modelling in the paper to explain what this figure is showing and describe why it does not match Figure 3. In addition, the fact that the figure contour scale only goes to 22 ug/m3 but the observations in Fig 2 reach over 60-100 ug/m3 would suggest that the model is not doing a good job at representing the plume. Given the efforts by other authors (e.g. Schmidt et al 2015) to demonstrate that their models provide good agreement with satellite column loading data before applying them to near surface concentrations, the lack of any model validation in this paper is particularly stark.

**RESPONSE:** The use of the EMEP4UK model was to illustrate the spatial distribution of the plume and to confirm the origins of the elevated sulfur species. The figure 3 presents the daily average $SO_2$ concentration compared to the hourly concentration measured at the surface in figure 2. To clarify how well the modelled matched the measured surface concentration an additional figure (Fig S3) has now been added to the supplementary information which provides information on the hourly resolution of

the EMEP4UK model compared to measurements. It has to be noted the reason the model does not always replicate the measurements is that the model output has a 50 x 50 Km² resolution. In addition other variables such as the injection height of the plume at source, frequently changing emission rate at source and as well as oxidation rates within the plume are unknown. Measured emissions rates from source would be required before a model vs observation study could be carried out and therefor beyond the scope of this paper. We have expand the text in section 3.1 in order to reflect our responses.

*"Supporting evidence that the ground-based measurements in September 2014 were picking up a volcanic signal is provided by the GOME2 instrument on the MetOp-B satellite, as it was able to track the $SO_2$ plume from the Holuhraun eruption site to the UK (Figure 3) on the 20th and 21st September. Modelling of the plume by EMEP4UK further confirmed the volcanic origin and dispersion of the observed $SO_2$ plumes both at Harwell and Auchencorth Moss (Figure 4) as did back trajectories using the HYSPLIT model, which can be found in figures S1-S2 in the supplementary material. To provide evidence that the EMEP4UK model was able to replicate the spatial distribution of the plume a comparison of the time series of observations to the model at both sites is given the supplementary material (figure S3). Though in general the EMEP4UK model does get the spatial distribution of the plume correct there are number of explanations why the model does not precisely replicate the plume compared to the measurements at the surface. The reasons include there a number of input variables which would have impacted the distribution of the plume including injection height, daily emission rates at source and oxidation rate within the plume, in addition the model output is 50Km x 50 Km resolution."*

Section 3.2: This is particularly interesting and one of the main new findings presented in this work. I have a few questions based on the text which the authors could hopefully easily incorporate the answers to. The fact that the difference in the plume aerosol diameter is so pronounced compared to the background is perhaps worth stating more clearly.
**RESPONSE:** We thank the reviewer that we did not highlight the differences in aerosol distribution clearly enough. We have added the following text:

*"As the plume passed over the sites, the presence of ultrafine particles also became pronounced, compared to the background atmosphere in the previous 24 hours."*

P6 Line 32 – the text talks about the slow oxidation of SO2 in the troposphere, are you referring to heterogeneous or homogeneous oxidation here or both?
**RESPONSE:** We thank the reviewer for pointing out that we had not clarified the reaction we were discussing. Therefor we have expand the text to provide additional information prior to presenting oxidation results.

*"$SO_2$ oxidation with the hydroxyl radical in the troposphere can be slow, taking up to two weeks under some conditions, though if $SO_2$ is taken up onto particles, oxidation rates are much faster, resulting in a lifetime of days or hours in clouds as $SO_2$ (von Glasow et al., 2009)."*

P6 Line 37 – you refer to the plume containing "young SO4", I wonder whether it would be more appropriate to say that it is a "young plume"? Do you have any idea of the travel time since emission? Does this fit with your "young" finding?

**RESPONSE:** We have reviewed the wording and reworded the text for this section, which includes the removal of the phrase "young $SO_4^{2-}$". See below:

*"In order to understand the oxidation of an $SO_2$ plume, Satsumabayashi et al. (2004) defined a sulfur conversion ratio ($F_s$) as $F_s = [PM_{2.5}\ SO_4^{2-}]/([SO_2]+ [PM_{2.5}\ SO_4^{2-}])$ (all concentrations in µg S m$^{-3}$), where a smaller value suggests a plume which has not undergone much atmospheric processing. The UK observatory datasets showed $F_s$ decreasing from ~1 (all S in the form of $SO_4^{2-}$) to $F_s$~0.2, (Figure 5) during the event implying that $SO_2$ oxidation had not had sufficient residence time (and oxidant exposure) to be complete."*

P7 Line 14-16: Is this true? This also sounds as though it needs consideration of the travel time. If the travel time was constant from the source to the observation point, then the particles arriving later in the day would have travelled longer in sunlight and so had a longer time to react. Perhaps it is just the wording of the sentence that needs tightening to make this clear. Should it be "with increasing time *after* sunrise"? And where you refer to "site" on line 16 do you mean the monitoring site or the eruption site?

**RESPONSE:** We are in agreement with the reviewer with what is being observed in figure 5 and that the particles arriving later in the day had been travelling in sunlight longer and therefor had more time to react. We have reworded the text to make this message clearer.

*"It is hypothesized that with increasing time after sunrise, the measurements at Auchencorth reflect particles whose nucleation was initiated further and further away from Auchencorth and had increasingly time to grow during transport."*

Section 3.3: This section is also one of the main findings and theories in this paper derived from the observations. It is an interesting conclusion, but also raised a number of questions in my mind that it would be useful for the authors to comment on in the paper. Firstly, whether the displacement has occurred due to the transport of the plume over the sea for such a long distance (and/or time) or whether this is a relatively local affect due to the site being not far inland. Second, if the plume had travelled directly south, it actually would have been over land for many miles before reaching the site, how would this fit the proposed mechanism?

**RESPONSE:**

Q1 and 2: We have highlighted that acid displacement events have already been demonstrated to occur at Auchencorth Moss in Twigg *et al* (2015) on occasions where high $NO_3^-$ has been observed and it is assumed the displacement is a combination of traversing the sea for a long distance as demonstrated in figure S1 (supplementary material) and the location of the site, which is close to the sea. We have added the following text to try and reflect this:

*"Further evidence of acid displacement was found at Auchencorth Moss when the ratio of $Na^+$ and $Cl^-$ was compared to the known ratio of sea water, where a large relative depletion of aerosol $Cl^-$ was found during elevated $SO_4^{2-}$, represented by a change in colour of the markers in figure 7. This is not the first time the site has observed acid displacement, Twigg et al. (2015) observed the site to be rich in sea salt due to its proximity to the sea, with 35% of the annual average of the inorganic composition of $PM_{2.5}$ attributed to sea salt. During high nitrate ($NO_3^-$) episodes it was observed on occasions that this coincided with an apparent depletion of $Cl^-$ from sea salt, which was attributed to the displacement of $Cl^-$ by $HNO_3$. "*

Third, does such a mechanism require transport to Scotland to have occurred near to the sea surface / within the boundary layer / or more local near-surface transport following above-BL transport over the ocean? Is there any data to support one of these over another?

**RESPONSE:** Background aerosol in the boundary layer and in the free troposphere can have sea salt aerosol present, so there is no requirement for the air mass to have been in the boundary layer, though aerosol concentrations tend to be higher in the boundary layer. We think the subtleties of this question could be best addressed with a modelling study where the aerosol loading and plume height could be varied. It is therefore beyond the scope of this study to answer the reviewers question but could be addressed in future work.

Conclusions: The first line states that the eruption perturbed \*all\* aspects of the UK atmosphere. As a first point this should be the atmospheric composition (not atmosphere, see technical corrections), but even so this seems to be rather overstating what has been presented. For instance there is no mention in the paper of changes to oxidant levels, impact on ammonium reactions, etc, which is understandable given the context, but would be necessary to justify the "all aspects" claim. Some minor, but careful, rewording of this sentence would bring it more in line with what has actually been presented.

**RESPONSE:** We accept the reviewers comment and have revised the phrasing to contain the following text:

*"The Holuhraun eruption perturbed the UK atmospheric composition periodically during the latter part of 2014."*

Technical Corrections:
**Response:** We apologise for the technical corrections and thank the reviewer for the time spent in identifying technical corrections.

p1 Line 25: "of the Holuhraun" needs modification to for example "of the Holuhraun fissure" or "at Holuhraun"

**RESPONSE:** Corrected

p1 Line 29: expand what EMEP stands for, or omit from sentence

**RESPONSE:** Corrected

p1 Line 34/35: missing "of" – "due to primary emissions of HCl"

**RESPONSE:** Corrected.

p2 Line 4: delete one of the two "were"

**RESPONSE: Corrected.**

p2 Line 8: add "of this type" to the end of the final sentence

**RESPONSE: Corrected**

p2 Line 22: add "is" to "but there is a very limited"

**RESPONSE: Corrected**

p2 Line 27 and p9 Line 30: the eruption was within the Bardarbunga volcanic system not the Holuhraun volcanic system. The eruption site and the eruption have been called Holuhraun.

**RESPONSE: Corrected to state:**
*"The recent Holuhraun eruption within the Bardarbunga volcanic system in Iceland.."*

P2 Line 29: recommend making "emission" plural, i.e. emissions

**RESPONSE: Corrected**

P2 Line 34: please explain what the EU-28 is/means for an international readership

**RESPONSE: We have edited the text.**

**"***The 28 countries European Union member countries (EU-28) total annual emissions of sulfur oxides….***"**

P3 Line 8: "Northern" should have a lower case "n"

**RESPONSE: Corrected**

P3 line 26: replace "for" with "of", i.e. "a detailed description of the instrument"

**RESPONSE: Corrected.**

P3 line 27: please expand the acronyms QA/QC

**RESPONSE: Corrected and the text now states: "***A detailed description of the instrument and quality assurance/ quality control (QA/QC) procedures …***"**

P3 line 27: replace "are" with "is", i.e. "by both instruments is given in"
**RESPONSE: Corrected.**

P3 line 28: need to add "the", i.e. "between the Auchencorth"

**RESPONSE: Corrected.**

P3 line 30: what is "IC"?

**RESPONSE: corrected now states : "...***ion chromatography (IC)...***"**

P3 line 30: need to add "a", i.e. "to achieve a lower detection"

**RESPONSE: Corrected**

P3 line 31: modify to be "therefore has an order of magnitude"

**RESPONSE: Corrected**

P4 line 26: a word is missing from "Downstream of is a gas"

**RESPONSE: Corrected now states: "***Downstream of the sampling train is a gas meter...***"**

P4 line 28: remove "this"

**RESPONSE: Corrected.**

P4 line 35: Gome should be capitalised, i.e. GOME

**RESPONSE: Corrected.**

P5 line 16: remove "below"

**RESPONSE: Corrected.**

P5 line 24: in a number of places in the text the authors use "high resolution analysis", this is not specific enough, I assume that they mean high temporal resolution not spatial? This should be included/made clear.

**RESPONSE: Corrected to state: ***"...high temporal resolution analysis (hourly measurements)..."***

P6 line 19: there are other references that could be included here for the observation of the plume (see General Comments). It would also be useful for the authors to clarify whether these observations occurred at approximately the same time (i.e. related to the same plume transport) or at different times during the prolonged eruption.

**RESPONSE:** We have added other reference which have already been dealt with in the general comments. Apologies that we did not clarify that the reported plumes elsewhere in Europe were observed at different periods during the eruption. We have now modified the text in order to highlight that different nations observed the plumes at different times.

*"The SO₂ plume was also observed across Ireland, Netherlands, Belgium and Austria (TS-2 in Supplementary Material of Gíslason et al. (2015)) during different periods of the fissure eruption."*

P7 line 4: change to "or 'banana' shape in Figure 5, starting with"

**RESPONSE: Corrected.**

P7 line 23-25: change line 23 to be "air quality impact from particulates during" and remove "due to particles" from line 25.

**RESPONSE: Corrected**

P7 line 35-36: The reference list is not needed here as these are already referred to or implied earlier in this sentence.

**RESPONSE: Corrected.**

P7 line 38: However, a reference is definitely needed for the molar ratio of HCl/SO2 being <1% near source.

**RESPONSE:** A reference has now been added.

P8 line 4: Is HCl correct at the end of this line? Should it be Cl-?

**RESPONSE: Apologies for the mistake. This has been corrected to state Cl⁻.**

P8 line 31: Add "at", i.e. "in particular at the sites", and South West should be lower case

**RESPONSE: Corrected.**

P8 line 32: add "in", i.e. "whereas in Northern Ireland", and change "were" to "was"

**RESPONSE: Corrected.**

P8 line 33: add "that", i.e. "noted, however, that there"

**RESPONSE: Corrected**

P8 line 37: suggest rephrasing "was not important to" to "was not significantly Different to normal" or similar

**RESPONSE:** Apologies, the words "annual sulfur deposition" were omitted. The sentence should have read:

*"The majority of the western UK received less than 20% of the long‑term average rainfall, hence the amount of sulfur deposited by wet deposition during this period was not important to the UK annual sulfur deposition budget and hence the environmental impact through acid deposition will have been minimal (Figure 9)."*

P9 line 4: change to "the UK atmospheric composition during the latter part. . ."

**RESPONSE: rephrased to state:**

*"The Holuhraun eruption perturbed the UK atmospheric composition periodically during the latter part of 2014."*

P9 line 5: change line to "Elevated SO2 was observed by the networks at both high and low temporal resolution. These observations complement the study by"

**RESPONSE: Corrected the requested phrasing.**

P9 line 8: remove the comma and change "to" to "in"

**RESPONSE: Corrected the phrasing.**

P9 line 10-11: I think we would expect particle formation and growth to be occurring in the plume based on past chemical and physical knowledge, so it would be better to say ". . . from the two EMEP supersites provide observational evidence for new particle formation and growth occurring as the plume. . ."

**RESPONSE:** Rephrased according to the suggestion.

*"…the two EMEP supersites provide observational evidence for new particle formation and growth was occurring as the plume passed over the UK."*

P9 line 14: add "work", i.e. "the recent modelling work undertaken"

**RESPONSE: Corrected.**

P9 line 22: add "that" to become "The study has highlighted that even though"

**RESPONSE: Corrected.**

P9 line 25: change "are" to "is"

**RESPONSE: Corrected.**

P9 Line 26: remove "the" from "concurrently with the SO2"

**RESPONSE: Corrected.**

Fig 1 caption: Repeat of "sites"

5  **RESPONSE: Corrected and rephrased.**

Fig 3 caption: explain what VC SO2 is and what DU is. Are these images snap-shots or aggregated daily totals or means? This needs to be stated.

10  **RESPONSE: The text has been updated:**

*"Observation of the volcanic plume from Iceland to and across the UK by the GOME2B satellite instrument, taken at the satellite overpass around 9:30 local time. GOME2B measures $SO_2$ column density, where VC is the vertical column, which is*
15  *the $SO_2$ concentration integrated vertically to provide a column density per unit surface area. $SO_2$ columns are given in Dobson Units (DU), the thickness the $SO_2$ layer would have at standard temperature and pressure in units of hundredths of a millimetre."*

Fig 5 caption: explain what the black line is
20
**RESPONSE:** We have modified the figure text to state that black line is the $F_s$.

Fig 7 caption: remove capitalisation from "Sea". It would be useful to explain the colouration of the dots in the main paper text and what this means for this event.
25
**RESPONSE:** Corrected. Have reword main manuscript text to state:

**"***Further evidence of acid displacement was found at Auchencorth Moss when the ratio of $Na^+$ and $Cl^-$ was compared to the known ratio of sea water, where a large*
30  *relative depletion of aerosol $Cl^-$ was found during elevated $SO_4^{2-}$, represented by a change in colour of the markers in Figure 7."*

Fig 9 caption: what are the orange lines?

35  **RESPONSE:** We have amended the figure caption, it now states:

[revised manuscript text omitted]

**Detection of the volcanic plume**

In order to further confirm the origins of the observed plume back trajectories were merged with the measured $SO_2$ data from the MARGA. The back trajectories and analysis was carried out using OpenAir software package (Carslaw, 2013), which calculates back trajectories with the HYSPLIT trajectory model (Hybrids Single Langrangian Integrated Trajectory Model, (Draxer and Hess, 1997)) using the global NOAA-NCEP/NCAR reanalysis data. For Auchencorth Moss the plume peaked on the 21/09/14 and is shown in FigS1 to originate from Iceland. The main trajectory, is over the highlands of Scotland, which does not have any known large sources of $SO_2$. At Harwell, the peak of the plume was on the 22/09/14 and again can be clearly seen to originate from Iceland (FigS2).

[Figure]

**Figure S1 96 hour back trajectories using the HYSPLIT model merged with SO₂ measurements from the MARGA instrument for the 21/09/14 to further demonstrate that peak SO₂ concentrations at Auchencorth Moss originated from the Holuhraun effusive eruption. (Figure produced using Open air; Carslaw and Ropkins, 2012)**

[Figure]

**Figure S2 96 hour back trajectories using the HYSPLIT model merged with SO₂ measurements from the MARGA instrument for the 22/09/14 to further demonstrate that peak SO₂ concentrations at Harwell originated from the Holuhraun effusive eruption (Figure produced using Open air; Carslaw and Ropkins, 2012).**

The EMEP4UK model was also used to confirm the distribution of the plume, as presented in Figure 4 in the main text. To provide further evidence of the agreement of the spatial distribution of the plume by the model, data for the sites Auchencorth Moss and Harwell data were plotted in a time series against the observed

concentrations. It is clear to see that the model is able to show that Auchencorth Moss the observed plume on the 21/09/14, however the site did not observe the plume predicated on the 22/09/14 or match the magnitude of the plume at the surface. At Harwell, the observed 3 pulses on consecutive days (21/09/14 - 23/09/14) from the volcanic plume were identified in the temporal pattern. There are a number of explanations why the magnitude and even the spatial of the distribution of the plume was not comparable to surface concentration measurements, these include that the emission rate from source and the injection height were variable, the model has a resolution of 50 x 50 Km$^2$ and so variations at surface are not well replicated at this spatial resolution.

[Figure]

**Figures S3 Times series of modelled plume by EMEP4UK and that observed by the MARGA instrument.**